# PromptAgent: Strategic Planning with Large Language Models Enables Expert-level Prompt Optimization

**Xinyuan Wang**[1][*] **Zhen Wang**[12][*][†] **Chenxi Li**[1][*]  **Fan Bai**[5]  **Haotian Luo**[2]
**Jiayou Zhang**[2]  **Nebojsa Jojic**[3]  **Eric Xing**[24]  **Zhiting Hu**[1]
[1]UC San Diego [4]Carnegie Mellon University
[3]Microsoft Research [5]Georgia Institute of Technology
[2]Mohamed bin Zayed University of Artificial Intelligence
{xiw136, zhw085, zhh019}@ucsd.edu

## Abstract

Highly effective, task-specific prompts are often heavily engineered by experts to integrate detailed instructions and domain insights based on a deep understanding of both instincts of large language models (LLMs) and the intricacies of the target task. However, automating the generation of such expert-level prompts remains elusive. Existing prompt optimization methods tend to overlook the depth of domain knowledge and struggle to efficiently explore the vast space of expert-level prompts. Addressing this, we present PromptAgent, an optimization method that autonomously crafts prompts equivalent in quality to those handcrafted by experts. At its core, PromptAgent views prompt optimization as a strategic planning problem and employs a principled planning algorithm, rooted in Monte Carlo tree search, to strategically navigate the expert-level prompt space. Inspired by human-like trial-and-error exploration, PromptAgent induces precise expert-level insights and in-depth instructions by reflecting on model errors and generating constructive error feedback. Such a novel framework allows the agent to iteratively examine intermediate prompts (states), refine them based on error feedbacks (actions), simulate future rewards, and search for high-reward paths leading to expert prompts. We apply PromptAgent to 12 tasks spanning three practical domains: BIG-Bench Hard (BBH), as well as domain-specific and general NLP tasks, showing it significantly outperforms strong Chain-of-Thought and recent prompt optimization baselines. Extensive analyses emphasize its capability to craft expert-level, detailed, and domain-insightful prompts with great efficiency and generalizability[1].

## 1 Introduction

Prompt engineering aims to craft effective prompts for harnessing the full potential of large language models (LLMs). Recent automatic prompt engineering, i.e., prompt optimization, has successfully studied training soft prompts (Lester et al., 2021; Hu et al., 2021; Wang et al., 2022), or searching for optimal combinations of discrete tokens (Shin et al., 2020; Deng et al., 2022; Zhang et al., 2022), by utilizing internal states or gradients of LLMs. For cutting-edge, proprietary API-based LLMs like GPT-4 (OpenAI, 2023b), prompt engineering largely relies on somewhat ad-hoc human-machine interactions. Human prompting experts thus need a unique blend of domain knowledge and intuition for LLMs to design the most effective prompts. For instance, an ideal prompt from human experts, shown in Figure 1, might integrate nuanced elements like task descriptions, domain knowledge, solution guidance, etc., all of which substantially boost prompt quality and performance. Automating expert-level prompt engineering on API-based LLMs presents significant challenges, largely due to the intricate nature of expert-level prompts, as illustrated in Figure 1. Although recent prompt optimization approaches have begun to utilize techniques like iterative sampling or

---

[*]Equal contribution

[†]Corresponding author

[1]Code and demo are available at: https://github.com/XinyuanWangCS/PromptAgent

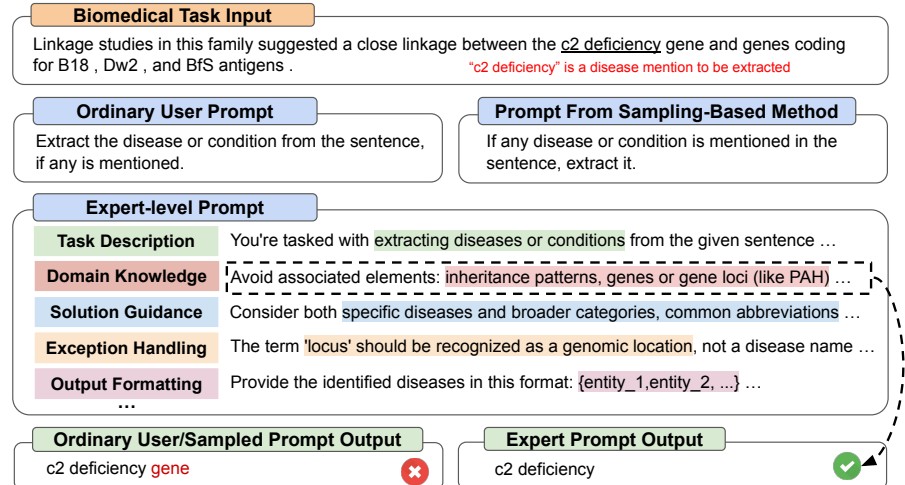

Figure 1: Expert-level prompt vs. ordinary human-written prompt and prompt from sampling-based methods (i.e., Automatic Prompt Engineer, Zhou et al. (2022)). The task is in the biomedical domain for extracting disease entities (NCBI, Doğan et al. (2014)). The expert prompt provides much richer domain-specific details and structured guidance than the other two, leading to the correct prediction.

evolutionary algorithms, such as Monte Carlo search (Zhou et al., 2022) or Gibbs sampling (Xu et al., 2023), they mostly employ heuristic methods like text edits or paraphrasing for generating candidate prompts (Zhou et al., 2022; Prasad et al., 2023). These approaches also often rely on straightforward iteration algorithms and lack a principled strategy to guide the exploration. Consequently, they tend to settle on local variants of prompts from ordinary users and rarely ascend to the excellence and nuances of expert-level prompts. Critically, many of these methods overlook that prompting engineering is essentially a human-in-the-loop application. In this process, humans refine prompts by fixing intermediate errors and integrating necessary domain knowledge through iterative interactions. This iterative refinement process characterizes the merits of how human experts craft superior prompts. Yet, the challenge remains that human exploration, while effective, can be expensive and less efficient at handling multiple errors simultaneously to explore the prompt space, thereby impeding the scalability of expert-level prompting.

To address the above challenges and combine human-like exploration with machine efficiency, we introduce PromptAgent in this paper. Drawing inspiration from human trial-and-error processes, PromptAgent seamlessly incorporates the principled planning approach, specifically Monte Carlo Tree Search (MCTS), to strategically optimize the prompting process. Notably, PromptAgent reformulates prompt optimization as a strategic planning problem to address the complexity of expert-level prompt space. Under this planning framework, it plays trial-and-error iteration to retrieve model errors and leverages the self-reflection ability of LLMs (Jang, 2023; Shinn et al., 2023; Pan et al., 2023) to generate insightful *error feedback*. This feedback, in turn, plays a critical role in effec-

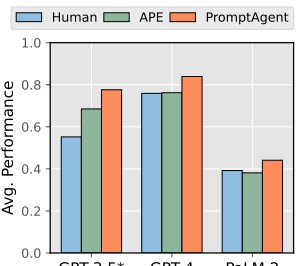

Figure 2: Prompt comparison across different base models.

tively inducing domain knowledge and guiding towards in-depth prompts. Through strategic planning, PromptAgent iteratively leverages insightful error feedback (action) to refine each version of prompts (state). Starting from an initial prompt (state), PromptAgent systematically grows the prompt space in a tree structure and prioritizes high-reward traces to navigate the vast space of expert-level prompts. The principled MCTS planning allows PromptAgent to look ahead and simulate future rewards, which are then backpropagated to update the beliefs about the current prompt so that PromptAgent can explore more promising alternatives later.

We demonstrate that PromptAgent can discover productive expert-level prompts by applying it to 12 tasks spanning three practical and distinct domains: BIG-Bench Hard (BBH) (Suzgun et al., 2022), as well as domain-specific and general NLP tasks. Starting with an initial human-written prompt and a small set of training samples, PromptAgent not only enhances the performance of the initial human prompt greatly but also significantly surpasses strong Chain-of-Thought (CoT) and recent prompt

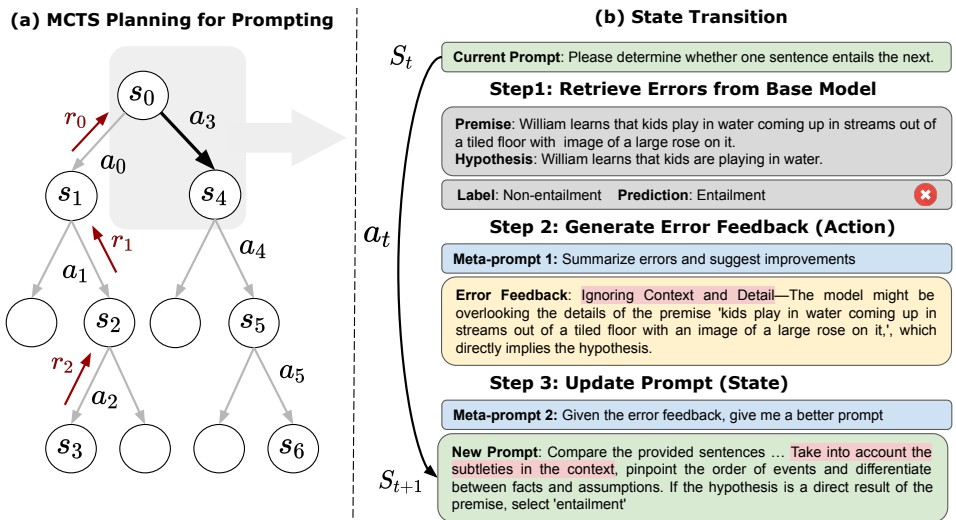

Figure 3: (a) MCTS (Monte Carlo Tree Search) planning for expert-level prompting. The tree structure enables strategic planning for PromptAgent. (b) A simplified state transition example. Given a current state (prompt), the base model (`gpt-3.5-turbo`) collects errors from the task dataset. The optimizer model (`gpt-4`) provides error feedback accordingly. The optimized model then updates the prompt according to the feedback and transits to the next state.

optimization baselines. For instance, Figure 2 shows PromptAgent consistently outperforms human and Automatic Prompt Engineer (APE) (Zhou et al., 2022) baselines across `GPT-3.5`, `GPT-4`, and `PaLM 2`, yielding improvements by 9.1%, 7.7% and 6% over APE, respectively. Extensive qualitative results further highlight the expert-level aspects of optimized prompts, indicating that PromptAgent effectively bridges the domain gap in challenging tasks, offering great exploration efficiency and generalizability. As we anticipate the emergence of even more powerful LLMs that can understand intricate instructions, we believe that expert-level prompting will spearhead the next era of prompt engineering, where PromptAgent stands as a pioneering step in this research direction.

## 2 METHODOLOGY

Given a base LLM $\mathcal{B}$ and a target task $\mathcal{T}$, the job at hand for a prompt engineer is to craft an optimized natural language prompt $\mathcal{P}^{\mathcal{T}}$ that maximizes the performance of $\mathcal{B}$ on $\mathcal{T}$. However, the gap between novice and expert prompt engineers can be significant, particularly for tasks demanding specialized domain expertise, such as in the biomedical domain. Our primary objective is to autonomously refine the task prompt $\mathcal{P}^{\mathcal{T}}$ to bridge this knowledge gap, minimizing human intervention. Most existing approaches rely on sampling local prompt alternatives iteratively, which is not only resource-intensive but also lacks assurance of yielding an optimal final prompt. In light of this, we introduce PromptAgent, an agent-based framework to produce expert-level task prompts via strategic planning and reflecting with error feedback during the prompting process, striking a proper balance of exploration and performance.

**Problem formulation.** Adopting a standard setting in prompt optimization (Zhou et al., 2022), we start with an initial natural language task prompt $\mathcal{P}_0$ (e.g., "Let's solve this problem step-by-step") and a small set of training samples from target task $\mathcal{T}$ as $(Q, A) = \{q_i, a_i\}_{i=1}^N$, where $q_i/a_i$ are input/output pairs for each sample (e.g., a question and its answer). Given the model input consisting of $\mathcal{P}$ and $q_i$, the base LLM $\mathcal{B}$ makes the prediction (typically through a left-to-right generation process) based on $p_{\mathcal{B}}(a_i|q_i, \mathcal{P})$[2]. The goal of prompt optimization is to find the optimal natural language prompt $\mathcal{P}^*$ that maximizes the performance towards a measure function $\mathcal{R}$ (e.g., accuracy). This can be formally defined as an optimization problem: $\mathcal{P}^* = \arg\max_{\mathcal{P} \in \mathcal{S}} \sum_i \mathcal{R}(p_{\mathcal{B}}(a_i|q_i, \mathcal{P}))$, where $\mathcal{S}$ denotes the sample space for a natural language prompt, an infinite and intractable space, if not impossible, to comprehensively enumerate. Conventionally, human experts draw upon a blend of heuristics and domain-specific insights to craft such prompts. Although previous optimization methods have attempted to leverage iterative sampling methods for prompt discovery (Zhou et al., 2022), we advance this line of research by proposing a unified framework that seamlessly integrates

---

[2]Note this is traditionally a zero-shot setting we focus on, where task prompt excludes any training samples.

strategic planning for superior, expert-level prompt optimization. Next, we introduce the formulation of PromptAgent and then present the planning-based prompt optimization.

## 2.1 PROMPTAGENT FRAMEWORK DESIGN

The goal of PromptAgent is to inject expert-level prior into the task prompt while maintaining efficient exploration via strategic planning. For planning, the state is represented by each version of task prompt $\mathcal{P}$, and the actions need to encapsulate the potential modifications over the current prompt. Each action could represent, for instance, editing certain words or phrases or paraphrasing $\mathcal{P}$ (Jiang et al., 2020; Prasad et al., 2023). However, to make the action space more meaningful and trigger prior knowledge, we instead propose a different action space. As shown in Figure 3 (b), the action is defined as an error feedback representing the direction of further refining the current prompt (state). Specifically, PromptAgent generates the action in two steps: collecting errors from training samples (Step 1) and reflecting on errors to draw useful lessons (Step 2) that potentially improve the prompt $\mathcal{P}$. Our action generation is also inspired by the recent self-reflection of LLMs (Pryzant et al., 2023).

Given the definition of state and action, PromptAgent formally models the prompt optimization as a Markov Decision Process (MDP) by the tuple $(\mathcal{S}, \mathcal{A}, T, r)$ in which $\mathcal{S}$ is the state space, $\mathcal{A}$ is the action space, $T$ is the transition function $T : \mathcal{S} \times \mathcal{A} \mapsto \mathcal{S}$, and $r$ is the reward function $r : \mathcal{S} \times \mathcal{A} \mapsto \mathbb{R}$. Given a current state $s_t$, PromptAgent generates an action $a_t$ based on $a_t \sim p_{\mathcal{O}}(a|s_t, m_1)$, where $m_1$ is a prompt working on another optimizer LLM $\mathcal{O}$ to help the action generation. The action generation is operationalized by Step 1&2 in Figure 3 (b). Then, PromptAgent obtains a new state based on the transition function $p_{\mathcal{O}}(s_{t+1}|s_t, a_t, m_2)$, where $m_2$ is another prompt helping the state transition and update the prompt, also working on the optimizer LLM $\mathcal{O}$. More specifically, given the action $a_t$ as the error feedback from Step 1&2, PromptAgent asks the optimizer to generate a new prompt considering previous states and error feedbacks. Finally, to assess the quality of each new state $s_t$ after applying action $a_t$, PromptAgent straightforwardly defines a reward function $r_t = r(s_t, a_t)$ as the performance on a held-out set separated from the given training samples. The exact definition of reward will depend on task-specific metrics.

## 2.2 STRATEGIC PLANNING FOR PROMPT OPTIMIZATION

The aforementioned reformulation of the prompt optimization enables us to seamlessly integrate PromptAgent with principle planning algorithms, notably the Monte Carlo Tree Search (MCTS). This enables strategically navigating the vast prompt space while balancing the exploration and exploitation in finding high-reward paths of error feedbacks, which leads to the most generalizable expert-level prompts. Specifically, we observe some error feedbacks (actions) may inject instance-specific details into task prompts (states) that are hard to generalize task-wise (exploitation), where we need strategic planning to explore novel error feedbacks for higher rewards (exploration). MCTS operationalizes such strategic planning, as shown in Figure 3 (a), by progressively constructing a tree structure with each node as a state and each edge as the action for transiting states. MCTS expands the tree strategically by maintaining a state-action value function, $Q : \mathcal{S} \times \mathcal{A} \mapsto \mathbb{R}$, which represents the potential future rewards for applying an action $a_t$ to a state $s_t$. In other words, we rely on this function, $Q(s_t, a_t)$, to look ahead and estimate the potential rewards for paths following the current state-action pair. To update this $Q$ function and expand the tree, MCTS iteratively performs four operations: *selection*, *expansion*, *simulation*, and *back-propagation*. The iteration process ends when a pre-defined number of iterations is reached, and we then select the highest-reward trace for the final prompt. We next explain the four operations in PromptAgent, and the pseudocode of our MCTS-based prompt optimization can be found in Algorithm 1 of the Appendix.

**Selection** is the first step that selects the most promising nodes at each level to be further expanded and explored. At each iteration, it starts from the root node $s_0$, traverses through each tree level, selects a subsequent child node at every level, and stops at a leaf node. When selecting the child node at each level, we leverage the *Upper Confidence bounds applied to Trees* (UCT) algorithm, which is well-known for balancing the exploitation (choosing high-value nodes) and exploration (choosing less-visited nodes) as follows:

$$a_t^* = \arg\max_{a_t' \in A(s_t)} \left( Q(s_t, a_t') + c \cdot \sqrt{\frac{\ln \mathcal{N}(s_t)}{\mathcal{N}(\mathrm{ch}(s_t, a_t'))}} \right) \tag{1}$$

where $A(s_t)$ is the action set for node $s_t$, $\mathcal{N}(s_t)$ is the number of visiting times for node $s_t$, $\mathrm{ch}(s, a)$ represents the child node for $s_t$ after applying action $a_t'$ and $c$ is a constant to adjust the exploration. As we can see, the first term signifies exploitation by the $Q$ value, and the second term indicates

exploration, measuring the uncertainty for less visited nodes. In other words, if a node was less explored and its child node was less visited before, the second term will be higher.

**Expansion** grows the tree by adding new child nodes to the leaf node reached by the previous *selection* step. This is done by applying the action generation and state transition (Figure 3 (b)) multiple times, resulting in multiple new actions and states. Note that we may sample multiple training batches to derive diverse error feedback (actions). Within new nodes, we then send the highest-reward one to the next *simulation* step.

**Simulation** is the lookahead step to simulate the future trajectories for the selected node from the previous *expansion* step. This step usually comes with a playout policy to reach the terminal state quickly and calculate the future rewards. The choice of playout could be flexible, such as choosing random moves until the terminal. To reduce the computation cost of simulation and simplify the process, we perform the previous *expansion* step iteratively until the terminal, i.e., we keep generating multiple actions and selecting the highest-reward node to proceed to the next tree level.

**Back-propagation** happens when a terminal state is met during the *simulation*. The terminal state is usually defined when a pre-defined maximum depth is reached, or an early-stopping criterion is encountered. We then back-propagate the future rewards along the path from the root to the terminal node by updating the $Q$ value function. Specifically, for each state-action pair in the path, $Q(s_t, a_t)$ is updated by aggregating the rewards from all future trajectories starting from $s_t$ as follows:

$$Q^*(s_t, a_t) = \frac{1}{M} \sum_{j=1}^{M} \left( \sum_{s' \in S_{s_t}^j, a' \in A_{a_t}^j} r(s', a') \right) \tag{2}$$

where $M$ is the number of future trajectories starting from $s_t$, $S_{s_t}^j$ and $A_{a_t}^j$ represent the $j$-th state and action sequences starting from $s_t$ and $a_t$, respectively.

PromptAgent executes the above four operations with a pre-defined number of iterations to stabilize the $Q$ values and fully grow the tree for exploring the vast prompt space. We finally need to select the best trace and node (i.e., prompt) for the final evaluation. Multiple alternative solutions can be leveraged for this output strategy, e.g., one could opt for the best node in the best path with the highest reward, or directly choose the leaf node with the largest number of visiting times. For simplicity and empirical purposes, we use the first strategy to select the output prompt, which works the best in our experiments.

## 3  EXPERIMENTS

**Tasks and Datasets.** We curate 12 tasks from three domains for thorough experiments: 6 *BIG-Bench Hard (BBH) (Suzgun et al., 2022)*; 3 domain-expert tasks, including NCBI (Doğan et al., 2014), Biosses (Soğancıoğlu et al., 2017), and Med QA (Jin et al., 2021); and 3 *general NLU* tasks, including TREC (Voorhees & Tice, 2000), Subj (Pang & Lee, 2004), and CB (De Marneffe et al., 2019). More task and dataset details are in Appendix A.1.

**Baselines and Implementations.** We compare PromptAgent with three types of baselines: human prompts, Chain-of-Thought (CoT) prompts, and recent prompt optimization methods: (1) *Human prompts* are human-designed instructions usually originating from the datasets. We have a few-shot version of human prompts with examples from Suzgun et al. (2022) for BBH tasks and randomly sampled examples from the training set for others. (2) *CoT prompts* are Chain-of-Thought prompts with reasoning steps. We use the CoT prompts from Suzgun et al. (2022) for BBH tasks and construct CoT prompts for other tasks. We uses "Let's think step by step" for *CoT (ZS)* (Kojima et al., 2022). (3) Prompt optimization methods include *GPT Agent* and *Automatic Prompt Engineer (APE)* (Zhou et al., 2022). GPT Agent represents the recent surge of interest in LLM-powered autonomous agents (Weng, 2023), such as Auto-GPT [3]. Such agents are expected to autonomously perform planning and self-reflection to solve most human requests, including optimizing task prompts. We then leverage one of the powerful ChatGPT Plugins (OpenAI, 2023a) with GPT-4, *AI Agents* [4] for prompt optimization. Lastly, APE (Zhou et al., 2022) is one of the most recent prompt optimization methods that proposes a Monte Carlo search-based method to iteratively propose and select

---

[3]https://github.com/Significant-Gravitas/AutoGPT
[4]https://aiagentslab.com/

Table 1: Prompting performance on BBH tasks. ZS: Zero-Shot, FS: Few-Shot. We select six challenging tasks from BBH (Suzgun et al., 2022), requiring domain knowledge (e.g., Geometry) or reasoning (e.g., Causal Judgement). Our method outperforms in 5/6 tasks, with only CoT surpassing in Object Counting. On average, our accuracy exceeds others by at least 9%.

| | Penguins | Geometry | Epistemic | Object Count. | Temporal | Causal Judge. | *Avg.* |
|---|---|---|---|---|---|---|---|
| Human (ZS) | 0.595 | 0.227 | 0.452 | 0.612 | 0.720 | 0.470 | 0.513 |
| Human (FS) | 0.595 | 0.315 | 0.556 | 0.534 | 0.408 | 0.620 | 0.505 |
| CoT (ZS) | 0.747 | 0.320 | 0.532 | 0.542 | 0.734 | 0.610 | 0.581 |
| CoT | 0.747 | 0.540 | 0.720 | **0.960** | 0.626 | 0.650 | 0.707 |
| GPT Agent | 0.696 | 0.445 | 0.406 | 0.502 | 0.794 | 0.520 | 0.561 |
| APE | 0.797 | 0.490 | 0.708 | 0.716 | 0.856 | 0.570 | 0.690 |
| PromptAgent | **0.873** | **0.670** | **0.806** | 0.860 | **0.934** | **0.670** | **0.802** |

Table 2: Prompt performance on specialized and general NLU tasks. Specialized tasks are three biomedical tasks explicitly asking for domain knowledge for prompting. General NLU tasks are used to demonstrate the generality of our method. Ours significantly outperformed in all tasks.

| | Domain-specific Tasks | | | | General NLU Tasks | | | |
|---|---|---|---|---|---|---|---|---|
| | NCBI (F1) | Biosses | Med QA | *Avg.* | Subj | TREC | CB | *Avg.* |
| Human (ZS) | 0.521 | 0.550 | 0.508 | 0.526 | 0.517 | 0.742 | 0.714 | 0.658 |
| Human (FS) | 0.447 | 0.625 | 0.492 | 0.521 | 0.740 | 0.742 | 0.429 | 0.637 |
| CoT (ZS) | 0.384 | 0.425 | 0.508 | 0.439 | 0.656 | 0.63 | 0.750 | 0.679 |
| CoT | 0.376 | 0.675 | 0.542 | 0.531 | 0.670 | 0.784 | 0.643 | 0.699 |
| GPT Agent | 0.125 | 0.625 | 0.468 | 0.406 | 0.554 | 0.736 | 0.339 | 0.543 |
| APE | 0.576 | 0.700 | 0.470 | 0.582 | 0.696 | 0.834 | 0.804 | 0.778 |
| PromptAgent | **0.645** | **0.750** | **0.570** | **0.655** | **0.806** | **0.886** | **0.911** | **0.868** |

prompts. More implementation details are in the Appendix: PromptAgent implementation is in A.3; the hyperparameter settings are in A.3, and the search baseline implementations are in A.4.

## 3.1 RESULTS AND ANALYSES

**Comparison with various prompting baselines.** Table 1 & 2 present a comprehensive comparison of expert-level prompts generated by PromptAgent against the baselines across 12 tasks. Observing BBH tasks from Table 1, PromptAgent significantly outperforms all baselines overall and achieves 28.9%, 9.5%, and 11.2% relative improvement over baselines - human prompts (ZS), CoT, and APE, respectively. It is noteworthy that CoT prompts are especially effective in BBH tasks than other tasks, similar to findings from Suzgun et al. (2022), because BBH tasks often require strictly formatted solutions that can be readily induced by the step-by-step CoT reasoning, e.g. *Object Counting*. However, PromptAgent still outperforms CoT by a great margin in all tasks (except *Object Counting*), indicating that our optimized expert-level prompt can lead to a bigger improvement over few-shot CoT reasoning (even under the zero-shot prompt setting). Regarding optimization methods, while GPT Agent has planning and self-reflection capability, its planning is only used for a single turn of prompt rewriting with limited exploration in prompt space. APE, on the other hand, shows a greater scale of searching ability, but its exploration is based on Monte Carlo search, without planning and error-based reflections. Both deficits of GPT Agent and APE suggest the necessity of strategic planning in PromptAgent to explore the prompt space and deliver expert-level prompts.

Table 2 presents results on domain-specific and general NLP tasks. The former encompasses a broad spectrum of biomedical tasks, such as information extraction, sentence similarity, and question answering. Crafting prompts for these tasks needs extensive domain knowledge and heavy LLM prompt engineering instincts, where straightforward human prompts and CoT prompts are ineffective. Iterative-sampling-based optimization methods, like APE, are promising to incorporate domain knowledge. However, PromptAgent surpasses APE significantly by +7.3% on average, suggesting PromptAgent can better induce effective domain knowledge to produce expert-level prompts and close the knowledge gap between novice and expert prompt engineers. For general NLP tasks, the efficacy and generality of PromptAgent are further emphasized, outperforming both CoT and APE by margins of +16.9% and +9%, respectively. This implies the nontrivial expert gap, even for general NLP tasks, underscoring the imperative for expert prompts in diverse applications.

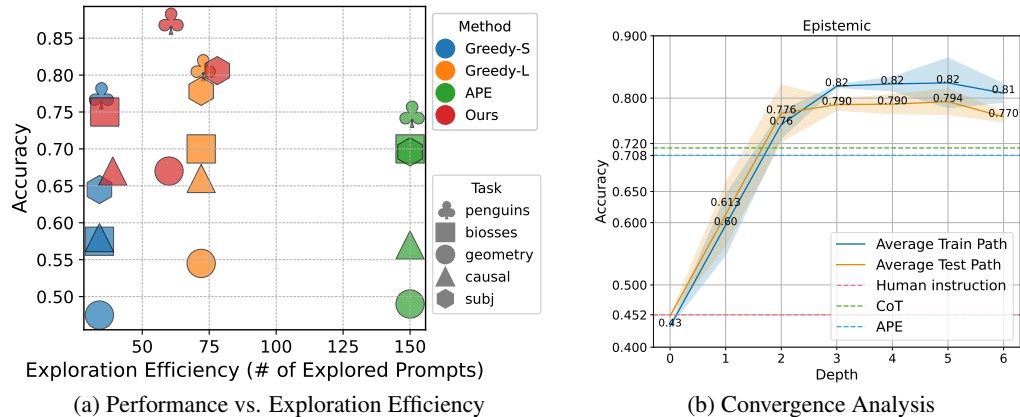

(a) Performance vs. Exploration Efficiency      (b) Convergence Analysis

Figure 4: (a) Exploration efficiency analysis. A proper balance of exploration and exploitation is crucial for search and planning. We compare the number of explored prompts between our method and three strong baselines. Ours achieves the best trade-off of performance and exploration (clustering in the top-left corner). (b) Convergence curves for *Epistemic* task. We visualize the mean and variance of the training and testing performance along the paths. We can observe that both curves increase at first and become stable after depth 3, suggesting a stable learning process

**Prompt generalization.** We further conduct experiments to investigate whether our optimized prompts can be generalized to other base LLMs. This emphasizes the robustness and transferability of expert-level prompts, which are urgently favorable and underpinning two key facts: (a) the domain insights and nuanced guidance in expert prompts can be seamlessly transferred across powerful LLMs, reinforcing the universal applicability of expert prompts, and (b) each task is optimized only once, leading to better computational efficiency. It is crucial to note that the primary goal of PromptAgent is to optimize prompts for state-of-the-art LLMs to achieve expert-level prompting, while less advanced and smaller LLMs, like GPT-2 or LLaMA, may not adeptly grasp the subtleties of these expert-level prompts, potentially causing significant performance drop. We evaluate our existing optimized prompt on one model more potent (GPT-4) and one less powerful (PaLM 2) compared to GPT-3.5. Figure 10 shows the aggregated results when we directly apply the optimized prompts from GPT-3.5 to GPT-4 and PaLM 2 (chat-bison-001) across all 12 tasks. We adopt the human and APE prompts as baselines. For certain tasks, we may employ slightly different prompts than those referenced in Table 1 to make PaLM 2 generate reasonable responses instead of persistent *null* answers. It is worth highlighting that when a stronger base LLM as GPT-4 is deployed, expert prompts manifest further enhancements, either on par with or outperforming Human and APE prompts in almost all tasks (11/12) (The only exception, *Temporal*, seems to be a solved task by GPT-4 with almost perfect accuracy). This underscores the untapped potential of expert prompting, especially with the evolution of LLMs in the future. When transferring expert prompts to a weaker LLM as PaLM 2, its performance drops dramatically across all tasks unsurprisingly. Nonetheless, we still observe PromptAgent exceeds both baselines on 7/12 tasks, with great improvements on domain-specialized tasks, such as *NCBI*, demonstrating the usefulness of domain insights from expert prompts. Detailed numbers can be found in Appendix Table 10.

**Ablation on search strategies.** To investigate the effect of strategic planning in PromptAgent systematically, we conduct an ablation study by comparing multiple alternative search strategies to MCTS, i.e., one-step Monte Carlo sampling (MC), depth-first greedy search (Greedy), and Beam search (Beam). Implementation details about search variants are specified in Appendix A.4. We use the same action generation and state transition as in PromptAgent and only replace the MCTS planning with each search method. We also restrict the three baselines to explore the same number of prompts. A subset of tasks is selected from the three domains to compare the above search variants. Table 9 shows that both Greedy and Beam greatly improve the MC baseline, suggesting the necessity of structured iterative exploration in our framework. When maintaining the same exploration efficiency, we observe comparable overall performance for Beam and Greedy. However, neither method strategically explores the prompt space since they operate in a strictly forward direction, lacking the capability to foresee future outcomes and backtrack to past decisions. In contrast, the strategic planning for MCTS allows PromptAgent to navigate complex expert prompt spaces more

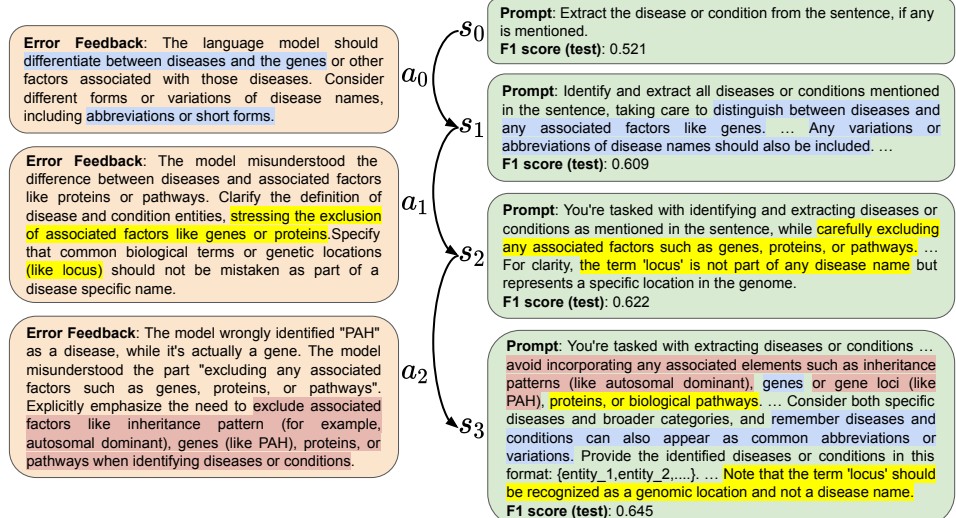

Figure 5: The MCTS state-action transition trajectory of the highest average reward path in NCBI. The initial state is $s_0$ with a human-written prompt. At each state transition step, a new prompt is crafted by adjusting the prior state based on error feedback. Highlighted colors indicate similar domain-specific insights. The last state integrates the information from the entire trajectory, elevating the F1 score from 0.521 to 0.645.

effectively, which significantly surpasses all search ablations on all tasks and gets a relative 5.6% overall improvement over the best baseline.

**Exploration efficiency analysis.** One of the key advantages of PromptAgent is that it can efficiently explore the prompt space via strategic planning. Exploration efficiency is also vital to make the computation cost of the search manageable. We thus analyze the exploration efficiency by comparing PromptAgent with some search baselines, including Greedy Search and APE from the previous section. Specifically, the exploration efficiency is measured by the number of prompts explored during the search. We plot its relationship with the task performance in Figure 4a. The Greedy-S and Greedy-L are Greedy Search with different explored prompts. The APE explores 150 prompts in each task. The figure shows that points of PromptAgent are clustered around the top left corner, indicating a superior performance with higher accuracy but fewer explored nodes (higher exploration efficiency). Notably, while increasing the number of prompts in Greedy Search may enhance performance, it demands higher exploration cost and still does not surpass PromptAgent. Also, without principled guidance, directionless searches like APE cannot effectively boost performance, even with larger exploration. Therefore, to maintain exploration efficiency and superior performance, strategic planning is crucial in PromptAgent and worthy of further research investment in future works. The detailed hyperparameter settings are in Appendix A.4.

**Convergence analysis.** To delve deeper into the learning process of PromptAgent, we examine the evolution of expert prompts throughout the tree planning process. We monitor and visualize performance changes with respect to tree depth. As illustrated in Figure 4b for the *Epistemic* task, we assess the performance across all nodes and aggregate both training and testing performance (reward) at each depth level. The plotted trajectories represent the evolution of average performance, illustrating a consistent improvement and gradually surpassing all baseline methods. Convergence plots for other tasks and hyperparameter settings are provided in Appendix E and Appendix A.3. A recurring pattern observed, similar to that in Figure 4b, indicates an upward trend in the initial iterations, suggesting a robust learning dynamic of PromptAgent to iteratively refine and enhance expert prompts.

**Qualitative analysis.** To further vividly show how PromptAgent iteratively converts error feedback (action) into better prompts (states), we conduct a qualitative analysis on the optimized trace from MCTS planning. As shown in Figure 5, we present the first four states and three action-state transitions of the best reward path for the NCBI task (Doğan et al., 2014). We highlight the domain insights by colors in both actions and states, where the same color indicates similar insights. As we can see, starting from a human-written prompt as $s_0$, PromptAgent proposes various insightful error

feedback (action) and effectively merges them into a new prompt (state) with improved performance. Specifically, throughout several transitions, the definition of a disease entity becomes increasingly refined, and task-specific details are integrated. The final state benefits from the accumulated information along the path, resulting in an expert prompt and an improved test F1 score.

**Additional ablation studies.** We conduct ablation studies of the hyperparameters including optimizer model, base model, exploration weight $c$, and iteration number in Appendix B

## 4 RELATED WORKS

**Prompt optimization**. Automatically discovering optimal prompts has emerged as a central challenge in the era of LLMs. For open-sourced LLMs, one can leverage their gradients to either train additional parameters, such as soft prompts (Li & Liang, 2021; Lester et al., 2021; Hu et al., 2021; Wang et al., 2022), or search for discrete prompts via gradient-based search (Shin et al., 2020; Wen et al., 2023) or reinforcement learning (Deng et al., 2022; Zhang et al., 2022). However, such methods are infeasible for closed-sourced LLMs, which requires gradient-free prompt optimization. Iteratively sampling and selecting is a typical method (Zhou et al., 2022). Numerous methods emphasize diversifying the prompt candidates - examples include edit-based methods like deleting or swapping phrases (Prasad et al., 2023), back translation (Xu et al., 2022), evolutionary operations (Guo et al., 2023; Fernando et al., 2023), or more relevantly, LLM rewriting based on natural language feedback (Zhou et al., 2022; Pryzant et al., 2023; Yang et al., 2023). There are also explorations into alternate sampling procedures like Monte Carlo search (Zhou et al., 2022), Gibbs sampling (Xu et al., 2023) or Beam search (Pryzant et al., 2023). Nevertheless, PromptAgent fundamentally differs from all the above methods in two ways. First, while primary search algorithms have been investigated (Zhou et al., 2022; Xu et al., 2023; Pryzant et al., 2023), we are the first to introduce strategic planning into prompting optimization research and incorporate fine-grained domain insights.

**Augmenting LLMs with self-reflection and planning**. Despite their remarkable capabilities, modern LLMs exhibit certain limitations, such as long-term coherence (Malkin et al., 2022), lacking an internal world model (Hao et al., 2023a), etc. Thus, augmenting LLMs has drawn extensive attention recently (Mialon et al., 2023; Ozturkler et al., 2022; Hao et al., 2023b; Jojic et al., 2023). Two common strategies are LLM self-reflection and planning. *Self-reflection* encourages the LLM to introspect its outputs and suggest more refined solutions (Jang, 2023; Pan et al., 2023). This has been leveraged to enhance a variety of applications, from complex computer tasks (Shinn et al., 2023), text generation (Welleck et al., 2022) to reasoning (Paul et al., 2023). *planning with LLMs* sheds light on enhancing these models, which is an essential ability for intelligent agents to take a sequence of actions in achieving specific goals (McCarthy et al., 1963; Bylander, 1994). One line of research is to prompt and evaluate LLMs on planning tasks directly (Liu et al., 2023). Another closer line of research is to augment the strategic reasoning ability of LLMs with planning-based algorithms, e.g. Tree of Thoughts (ToT) applies DFS/BFS, while both CoRe (Zhu et al., 2022) and RAP (Hao et al., 2023a) utilize MCTS to navigate richer reasoning paths. Yet, in contrast to existing endeavors in LLM augmentation, PromptAgent is the first novel framework for synergistically marrying the spirits of self-reflection and planning specifically tailored for prompt optimization.

## 5 CONCLUSION

In this paper, we introduce PromptAgent, a novel prompt optimization framework capable of autonomously crafting expert-level prompts for a given task. Expert-level prompting distinguishes itself from traditional prompt engineering by its effectiveness of seamlessly integrating domain insights and closing the knowledge gap for domain experts. To achieve this, central to PromptAgent is the novel perspective of viewing prompt optimization as a strategic planning problem, leveraging the power of MCTS planning to strategically and efficiently traverse the complex prompt space. PromptAgent incorporates domain-specific knowledge from tasks into the newly generated prompts through a trial-and-error manner based on the self-reflection abilities of LLMs. We tested the PromptAgent on 12 diverse tasks spanning three distinct domains. The prompts optimized by PromptAgent consistently exhibited expert-level characteristics, enriched with domain-specific details and guidance. These prompts significantly outperformed both human-written, Chain-of-Thought prompting and other optimized method baselines. Further in-depth analyses revealed superior transferability, exploration efficiency, and quality for our expert prompts, paving the way for future prompt engineering to unlock the sophisticated task understanding of state-of-the-art LLMs.

LIMITATION DISCUSSION

While PromptAgent demonstrates significant promise in optimizing expert prompts across a diverse range of tasks, its application to highly specialized domains, particularly where domain knowledge of the optimizer is limited (Bhayana et al., 2023; Azizi et al., 2023), presents unique challenges. Our current implementation, employing GPT-4 as the optimizer, effectively addresses a wide array of tasks, including those in the BIG-bench, biomedical, and NLP domains. However, the scope and depth of knowledge within LLMs, like GPT-4, are inherently tied to the state of LLMs and the data they have been pre-trained on (Yin et al., 2023). This raises concerns in domains where data is sparse or protected, such as in healthcare under HIPAA regulations [5].

While addressing these limitations is beyond the current paper's scope, we propose a range of strategies to enhance the adaptability of PromptAgent in domains where employing GPT-4 as the optimizer LLM may not be ideal. These include but are not limited to, adapting GPT-4 in specialized domains using highly tailored knowledge prompts to better trigger domain knowledge; leveraging retrieval techniques to supplement domain knowledge; implementing a quality control mechanism for error feedback; integrating a hybrid optimizer that combines the broad capabilities of general LLMs with specialized domain LLMs; and engaging with human experts to provide domain-specific guidance in the optimization process. Moreover, data augmentation techniques using synthetic or anonymized datasets, particularly for sensitive domains, are also promising to enrich the domain expertise of LLMs while maintaining compliance with privacy standards. Future research should focus on refining these strategies to mitigate the current limitations, broadening the scope and impact of PromptAgent in the evolving landscape of expert-level prompt applications.

ACKNOWLEDGMENTS

We are grateful to the anonymous reviewers for their constructive comments and suggestions. We are grateful to Shibo Hao, Yi Gu, Qiyue Gao, and other members of UCSD MixLab for their valuable discussions. We also thank Weijia Xu for discussing the baseline details of Reprompting (Xu et al., 2023).

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

# A IMPLEMENTATION DETAILS

## A.1 TASKS AND DATASETS.

To fully demonstrate the expert-level prompt optimization of PromptAgent, we curate 12 tasks from three domains for thorough experiments: *BIG-Bench Hard (BBH)*, domain-expert, and *general NLU* tasks. BBH tasks (Suzgun et al., 2022) are a set of challenging BIG-Bench tasks (Srivastava et al., 2023) that are beyond the capabilities of current LLMs, in which we select 6 BBH tasks based on their requirement for domain knowledge (i.e., geometric shapes, epistemic, and causal judgment) and multi-step reasoning (i.e., penguins in a table, object counting and temporal sequences). Moreover, we select three domain-specific tasks in the biomedical domain, which can explicitly benefit from expert-level prompting, including NCBI (Doğan et al., 2014), Biosses (Soğancıoğlu et al., 2017), and Med QA (Jin et al., 2021). Last but not least, to show that PromptAgent can further improve traditional NLP tasks, we select three well-known NLU tasks, i.e., TREC (Voorhees & Tice, 2000), Subj (Pang & Lee, 2004), and CB (De Marneffe et al., 2019).

Table 3: Data split

| Task | Train | Test |
|---|---|---|
| **Bigbench** | | |
| Penguins | 70 | 79 |
| Geometry | 150 | 200 |
| Epistemic | 500 | 500 |
| Object counting | 300 | 500 |
| Temporal | 300 | 500 |
| causal judgement | 90 | 100 |
| **Domain Knowledge** | | |
| NCBI | 2000 | 940 |
| Biosses | 60 | 40 |
| Med QA | 2000 | 500 |
| **General NLP** | | |
| Subj | 400 | 1000 |
| TREC | 400 | 500 |
| CB | 125 | 56 |

**Dataset Split.** For datasets with predefined testing sets, we use them directly as our testing set. However, if the set exceeds 1000 examples, we limit it to the top 1000. If no default testing set is provided, we shuffle the data and allocate approximately half for testing purposes. The remaining data forms the training set, from which batches are sampled. Additionally, a smaller subset is reserved for reward calculation. The size of this subset is dictated by the size of the training set, typically ranging between 60 to 200 examples, with a default of 150 if the training set is larger than 150 examples. The data split is in Table 3.

## A.2 INPUT FORMULATION

We form our model input in the following format:

Prompt + Task Prefix + Question + Task Suffix + Answer Format

The "Prompt" is our optimization target; the "Task Prefix" (Optional) is the task-specific background intro (For example, a table of background data in the Penguins); the "Question" is the main body of the task's question; the "Task Suffix" (Optional) includes the options (For example, yes/no, entailment/non-entailment, or A, B... in tasks with multiple choices); and the "Answer Format" (Optional) is designed for answer caption from the model's response. The input format is in Table 11.

## A.3 PROMPTAGENT IMPLEMENTATION DETAILS

**PromptAgent (Ours).** PromptAgent performs MCTS planning within the prompt space, requiring both terminal state conditions and a reward function. A terminal state is achieved when the path length hits *depth_limit*. The reward function is determined by the base model's performance on the held-out set. For computational efficiency to avoid unnecessary exploration, we also apply an early-stopping method after depth is larger than 2: if the state's reward is less than a *min_threshold* or larger than a *max_threshold*, we then reach an early-stopping state. Specifically, *min_threshold* is the average of the rewards of its parent node and the root node, while *max_threshold* is the maximum of all the current nodes, encouraging shorter paths.

1. **Initialization**. The PromptAgent-MCTS algorithm starts with an initial prompt as the root node. For BBH tasks, we directly adopt the task "description" from the original datasets

as the initial prompts, except that *Object Counting*'s default description doesn't follow the format of instruction. We crafted the initial prompts for the rest of the tasks according to their task objectives or question-answer formats. The root node will be evaluated to obtain the reward before the first expansion.

2. **MCTS Iterations**. The agent will perform 12 MCTS iterations. During the selection step, starting from the root node, the best child node will be added to the path according to its UCT value (Equation 1), and the exploration weight $c$ in UCT is 2.5. During the expansion step, *expand_width* batches (*batch_size* is 5) of examples will be sampled from the training set, and each batch will be fed to the base model to collect the errors. If there is no error, this sample-forward loop will iterate until an error is found. The errors will be formatted using **error_string** (illustrated in Table 11) and inserted into **error_feedback** (illustrated in Table 11, Meta-prompt 1 in Figure 3) to summarize errors by the optimizer. **state_transit** prompt (illustrated in Table 11, Meta-prompt 2 in Figure 3) contains the expanding node's prompt, the trajectory of prompts (list of prompts from the root of the expanding node on the currently selected path), and the error summarization, which is fed into the optimizer to generate *num_samples* new prompts (nodes). The new nodes will be evaluated and added as the expanding node's children if they are not terminal nodes. Each expansion will generate *expand_width* × *num_samples* new prompts. The simulation step will recursively expand the last node in the path and pick the one with the highest reward to add to the path. When the last node satisfies the terminal condition or early-stopping condition, the simulation is stopped. During the back-propagation, from the last node to the root, the cumulative rewards (the sum of rewards from the node to the leaf/terminal node) will be appended to the node's cumulative reward list, the average of which will be the node's $Q$ (Equation 2). We have three hyperparameter settings: Standard, Wide, and Lite in Table 4. In the Standard and Lite experiments, both have an *expand_width* of 3 and *num_samples* of 1, but their *depth_limit* are 8 for Standard and 4 for Lite. Wide experiment has *expand_width* is 3 and $num\_samples = 2$ to generate more nodes in each expansion step, but with a *depth_limit* of 6 to limit the total number of explored prompts. We select the best setting for each task based on the final rewards.

3. **Output strategy**. Each MCTS iteration will output one path from the root node to the leaf node, and there are tens of nodes generated after the searching process. We select the path with the highest average reward, then pick the prompt with the highest reward in the path as the final output prompt. We employ this strategy because the path with the highest average reward represents the best overall search trajectory. Also, the best prompt might not always be the last node on the optimal path, given that it may be a terminal state by reaching the depth limit.

Table 4: Hyperparameter settings for PromptAgent Experiments

| Experiment Name | Standard | Wide | Lite |
|---|---|---|---|
| *depth_limit* | 8 | 6 | 4 |
| *expand_width* | 3 | 3 | 3 |
| *num_samples* | 1 | 2 | 1 |

**Hyperparameters and default settings.** For all datasets, we use the original testing set by default if publicly available; otherwise, we use the development set for testing. If there is no official training/testing split, such as BBH tasks, we sample a reasonably large set for stable testing. As stated in Section 2.1, we also split a portion of training samples for calculating the reward. The details of the dataset split can be found in Appendix Table 3. Unless further specified, for the base LLM to be optimized, we target the decently powerful one and use GPT-3.5 as the default base LLM. For the optimizer LLM, we need one with a good self-reflection ability and use GPT-4 as the default optimizer LLM. We set the temperature as 0.0 for base LLM to make predictions and 1.0 in other contexts. When implementing PromptAgent, we set the number of iterations for MCTS $\tau$ as 12, and the exploration weight $c$ in Equation 1 as 2.5. During the expansion step, we generate an action based on the errors from a batch of training samples (batch size is 5). We sample 3 batches, and for each batch, we generate multiple new prompts.

## A.4 BASELINES IMPLEMENTATION DETAILS

We illustrate the details for various baselines in our experiments.

**Monte Carlo (MC).** MC performs one-step sampling multiple times and selects the best one as the optimized prompt. It uses the same prompt sampling method as PromptAgent, but limits the searching depth to one. In the search ablation study, we sampled 72 new prompts in each task.

**Beam Search (Beam).** Beam also uses the same expand function as PromptAgent. Each node, except the root, will be expanded into 3 new nodes, and the beam width is 3, meaning that there will be 9 nodes in each depth of the search tree, and the best 3 nodes will be kept for the next expansion. The root will be expanded into 9 new nodes. The search depth is 8, so there will be 72 nodes or new prompts in total.

**Greedy Search (Greedy).** Greedy is based on the Beam Search, but the beam width is one, so the algorithm turns into a depth-first greedy search. We conducted 2 experiments, Greedy-S and Greedy-L, in Figure 4a, with the same search depth of 8 but different expand widths. The Greedy-S's expand width is 3, with 34 prompts in total. The Greedy-L has an *expand_width* of 9 and 72 nodes in total, which is also referred to as the Greedy baseline in Table 9.

**APE (Zhou et al., 2022).** We employ the iterative APE with one iteration as our baseline, as suggested by the original paper (Zhou et al., 2022). When generating new prompts, a mini-batch comprising 5 data pieces is sampled as Input-Output examples for APE. Specifically, for **Initial Proposal Step**, by default, 10 data batches are sampled, with each batch being used to generate 10 new prompts. This results in a total of 100 candidate prompts during the initial step. (Due to the longer processing time of Med QA, only 25 candidates are generated for it in this phase.) Subsequently, the five prompts with the highest evaluation scores are chosen for the iterative proposal step. For **Iterative Proposal Step**, similar to the initial phase, 10 batches of data are sampled for each proposed prompt, resulting in a total of 50 candidate prompts in this step. Following this, the prompt with the top evaluation score is chosen as the optimized prompt.

## A.5 MCTS ALGORITHM FOR PROMPTAGENT

---

**Algorithm 1** PromptAgent-MCTS($s_0, p_\theta, r_\theta, p_\phi, d, L, \tau, c$)

---

**Inputs:**
 Initial prompt (state) $s_0$, state transition function $p_\theta$, reward function $r_\theta$, action generation function $p_\phi$,
 number of generated actions $d$, depth limit $L$, iteration number $\tau$, exploration weight $c$ (Equation 1)
**Initialize:**
 State to action mapping $A : \mathcal{S} \mapsto \mathcal{A}$, children mapping ch $: \mathcal{S} \times \mathcal{A} \mapsto \mathcal{S}$, rewards $r : \mathcal{S} \times \mathcal{A} \mapsto \mathbb{R}$,
 State-action value function $Q : \mathcal{S} \times \mathcal{A} \mapsto \mathbb{R}$, visit-time counter $\mathcal{N} : \mathcal{S} \mapsto \mathbb{N}$
**for** $n \leftarrow 0, \dots, \tau - 1$ **do**
  **for** $t \leftarrow 0, \dots, L - 1$ **do**
    **if** $A(s_t)$ is not empty **then**                               ▷ selection
      $a_t \leftarrow \arg\max_{a \in A(s_t)} \left( Q(s_t, a) + c \cdot \sqrt{\frac{\ln \mathcal{N}(s_t)}{\mathcal{N}(\text{ch}(s_t, a))}} \right)$
      $s_{t+1} \leftarrow \text{ch}(s_t, a_t), r_t \leftarrow r(s_t, a_t), \mathcal{N}(s_t) \leftarrow \mathcal{N}(s_t) + 1$
    **else**                                          ▷ expansion and simulation
      **for** $i \leftarrow 1, \dots, d$ **do**
        Sample $a_t^i \sim p_\phi(a|s_t)$, $s_{t+1}^i \sim p_\theta(s|s_t, a_t^i)$, and $r_t^i \leftarrow r_\theta(s_t, a_t^i)$
        Update $A(s_t) \leftarrow \{a_t^i\}_{i=1}^d$, $\text{ch}(s_t, a_t^i) \leftarrow s_{t+1}^i$, and $r(s_t, a_t^i) \leftarrow r_t^i$
      **end for**
      $a_t \leftarrow \arg\max_{a_t^i \in A(s_t)} r_t^i(s_t, a_t^i)$
      $s_{t+1} \leftarrow \text{ch}(s_t, a_t), r_t \leftarrow r(s_t, a_t), \mathcal{N}(s_t) \leftarrow \mathcal{N}(s_t) + 1$
    **end if**
    **if** $s_{t+1}$ is an early-stopping state **then break**
  **end for**
  $T \leftarrow$ the actual number of steps
  **for** $t \leftarrow T - 1, \dots, 0$ **do**                          ▷ back-propagation
    Update $Q(s_t, a_t)$ with $\{r_t, r_{t+1}, \dots, r_L\}$ based on Equation 2
  **end for**
**end for**

---

# B  ADDITIONAL EXPERIMENT RESULTS

## B.1  ABLATION EXPERIMENT OF OPTIMIZER MODEL

Table 5 shows the results on two biomedical tasks and one general NLU task of ablating the optimizer LLM with GPT-3.5 and GPT-4. Note that the default base model is GPT-3.5. The first row directly prompts GPT-3.5 with the initial prompt. The second and third rows use GPT-3.5 and GPT-4 for the optimizer LLM, respectively. As we can see, when replacing GPT-4 with GPT-3.5, which contains much less domain knowledge, we still obtained substantial gains over the initial prompt baseline. While the performance is lower than GPT-4 as expected, the drop is tolerable and marginal, especially in the domain-specific task, NCBI, with only about 3 point drop. Such results indicate the robustness of the overall PromptAgent framework to strategically search for high-quality domain insights, instead of solely relying on the knowledge of optimizer LLM.

Table 5: Ablation study of optimizer model: Prompt performances of using both GPT-3.5 as base and optimizer LLM (second row), and using GPT-3.5 as base model and GPT-4 as optimizer (third row).

| Optimizer | NCBI (F1) | Biosses (Acc.) | Subjective (Acc.) |
|---|---|---|---|
| Initial Prompt | 0.521 | 0.550 | 0.517 |
| GPT-3.5 | 0.615 | 0.675 | 0.747 |
| GPT-4 | **0.645** | **0.750** | **0.806** |

## B.2  ABLATION EXPERIMENT OF BASE MODEL

The choice of the base model is very flexible in the PromptAgent framework, which could be GPT-3.5, GPT-4, PaLM, or other LLMs. However, using a weaker LLM as the base model, such as PaLM 2 (chat-bison-001), could lead to a performance drop, partially due to the weaker understanding of expert prompts. This can be indirectly observed from the generalization experiment, where we transfer GPT-3.5-based prompts to PaLM 2 with a performance drop. We conduct new experiments in Table 6 to directly observe the effect of using PaLM 2 as the base model. As we mentioned in the paper, PaLM 2 is weaker than GPT-3.5, but we can see PromptAgent can still improve its performance greatly. Table 6 shows the results in three tasks using different base models, PaLM 2 and GPT-3.5, both against GPT-4 as the optimizer. The first row shows the worse initial performance using PaLM 2, indicating its weaker understanding of human prompts. After using PromptAgent to optimize PaLM 2 in the second row, we observe significant improvement over the first row, indicating the effect of PromptAgent to optimize weaker LLMs.

Table 6: Ablation study of base model.

| Prompt and Base Model | Penguins | Biosses | CB |
|---|---|---|---|
| Initial Prompt (PaLM2) | 0.215 | 0.500 | 0.571 |
| Optimized Prompt (PaLM2) | 0.443 | 0.600 | 0.839 |
| Initial Prompt (GPT-3.5) | 0.595 | 0.550 | 0.714 |
| optimized Prompt (GPT-3.5) | **0.873** | **0.750** | **0.911** |

## B.3  ABLATION EXPERIMENT OF EXPLORATION WEIGHT

**Exploration weight** $c$: In the paper, we choose $c = 2.5$ in Equation 1 because the formula of the exploration rate is $c \cdot \sqrt{\ln \mathcal{N}(s_t)/\mathcal{N}(\text{ch}(s_t, a'_t))}$, so we set some values of the visited times of parent node $\mathcal{N}(s_t)$ and child node $\mathcal{N}(\text{ch}(s_t, a'_t))$. We find $c = 2.5$ can make the change of this value at a reasonable scale, when the $\mathcal{N}(s_t)$ and $\mathcal{N}(\text{ch}(s_t, a'_t))$ increase. We also show new experiments in Table 7, showing that when $c$ is small (1.0), the MCTS will only have one path, while $c$ is large (4.0), the MCTS will almost generate a new path in each iteration. Both of them are out of balance; $c$ is small, so the prompt space is not well explored; while $c$ is large, each path won't be visited multiple times to update the reward; we thus use $c = 2.5$ as a reasonable setting.

Table 7: Ablation study of exploration weight $c$: Weight for balancing exploration vs. exploitation.

| $c$ | CB | Biosses | Penguins |
|---|---|---|---|
| 1.0 | 0.768 (27 nodes) | 0.650 (30 nodes) | 0.759 (27 nodes) |
| 2.5 | **0.911** (45 nodes) | **0.750** (36 nodes) | **0.700** (55 nodes) |
| 4.0 | 0.857 (66 nodes) | 0.700 (55 nodes) | 0.835 (75 nodes) |

## B.4 ABLATION EXPERIMENT OF ITERATION NUMBER

Like many other empirical studies, we select hyperparameters based on intuitions and preliminary experiments. We now show additional hyperparameter selection experiments on iteration number and the exploration weight $c$.

**Iteration**: In the paper, we select the iteration based on our experience that a small number of iterations can't explore the prompt space well, but too many iterations are not necessary because they may have a potential problem of overfitting on the training set. We add new experiments in Table 8 to test various choices of the iteration number, including iterations of 8, 12, and 16. The iteration of 8 is too small, which leads to a drop in performance, and the iteration of 16 is too large for the overfitting issue, leading to a marginal drop.

Table 8: Ablation study of iteration number of MCTS.

| Iteration | CB | Biosses | Penguins |
|---|---|---|---|
| 8 | 0.804 | 0.625 | 0.722 |
| 12 | 0.911 | 0.750 | 0.873 |
| 16 | 0.893 | 0.725 | 0.835 |

## B.5 ABLATION EXPERIMENT OF SEARCH METHODS

Table 9: Ablation study on search methods. MC: Monte Carlo search, Greedy: greedy depth-first search, Beam: beam search. Testing tasks are representative of three task domains from BBH (Suzgun et al., 2022), domain-expert, and general NLU. Our method consistently outperforms all other ablated search algorithms across every task we evaluated.

| | MC | Beam | Greedy | MCTS (Ours) |
|---|---|---|---|---|
| Penguins | 0.772 | 0.823 | 0.810 | **0.873** |
| Biosses | 0.575 | 0.675 | 0.700 | **0.750** |
| Geometry | 0.490 | 0.610 | 0.545 | **0.670** |
| Causal | 0.650 | 0.610 | 0.660 | **0.670** |
| Subj | 0.692 | 0.765 | 0.778 | **0.806** |
| *Average* | 0.635 | 0.697 | 0.698 | **0.754** |

## B.6 DETAILED PERFORMANCE COMPARISON ON PROMPT GENERALIZATION EXPERIMENTS

Table 10: Prompt generalization results. While we optimize GPT-3.5 as the default base LLM, its optimized prompts are transferable to other base LLMs like GPT-4 and PaLM 2 (chat-bison-001). GPT-4 sees further enhancement with our prompts, beating baselines in 11/12 tasks. Weaker LLMs like PaLM 2 may have challenges with our advanced prompts but still surpass baselines in 7/12 tasks. Overall, ours can significantly beat baselines with different base LLMs.

| | GPT-3.5 | | | GPT-4 | | | PaLM 2 | | |
|---|---|---|---|---|---|---|---|---|---|
| Tasks | Human | APE | Ours | Human | APE | Ours | Human | APE | Ours |
| Penguins | 0.595 | 0.747 | **0.797** | 0.772 | 0.848 | **0.962** | 0.430 | 0.443 | **0.456** |
| Geometry | 0.227 | 0.490 | **0.670** | 0.495 | 0.445 | **0.680** | 0.290 | 0.215 | **0.360** |
| Epistemic | 0.452 | 0.708 | **0.806** | 0.734 | **0.848** | **0.848** | 0.470 | 0.392 | **0.588** |
| Object Count. | 0.612 | 0.716 | **0.860** | 0.830 | 0.852 | **0.888** | 0.290 | **0.378** | 0.320 |
| Temporal | 0.720 | 0.856 | **0.934** | 0.980 | **0.992** | 0.982 | 0.540 | 0.522 | **0.620** |
| Causal Judge. | 0.470 | 0.570 | **0.670** | 0.740 | 0.740 | **0.770** | **0.440** | **0.440** | 0.430 |
| NCBI (F1) | 0.521 | 0.576 | **0.645** | 0.588 | 0.428 | **0.697** | 0.016 | 0.025 | **0.177** |
| Biosses | 0.550 | 0.700 | **0.750** | 0.700 | 0.775 | **0.800** | 0.500 | 0.300 | **0.600** |
| Med QA | 0.508 | 0.470 | **0.570** | 0.770 | 0.758 | **0.774** | **0.284** | 0.274 | 0.276 |
| Subj | 0.517 | 0.696 | **0.806** | 0.867 | 0.805 | **0.879** | 0.496 | **0.537** | 0.499 |
| TREC | 0.742 | 0.834 | **0.886** | 0.716 | 0.764 | **0.876** | 0.380 | **0.400** | 0.230 |
| CB | 0.714 | 0.804 | **0.914** | **0.911** | 0.893 | **0.911** | 0.571 | 0.643 | **0.732** |
| *Average* | 0.552 | 0.685 | **0.776** | 0.759 | 0.762 | **0.839** | 0.392 | 0.381 | **0.441** |

## C  TASK INPUT AND META PROMPT FORMAT

In this section, we will demonstrate the Meta formats and some model input examples.

Table 11: Meta Formats.

| Format Name | Meta Format |
|---|---|
| input_format | {prompt}
{task_prefix}
{question}
{task_suffix}
{answer_format} |
| error_string | <{index}>
The model's input is:
{question}

The model's response is:
{response}

The correct label is: {label}
The model's prediction is {prediction} |
| error_feedback | I'm writing prompts for a language model designed for a task.

My current prompt is:
{cur_prompt}

But this prompt gets the following examples wrong:
{error_string}

For each wrong example, carefully examine each question and wrong answer step by step, provide comprehensive and different reasons why the prompt leads to the wrong answer. At last, based on all these reasons, summarize and list all the aspects that can improve the prompt. |
| state_transit | I'm writing prompts for a language model designed for a task.

My current prompt is:
{cur_prompt}

But this prompt gets the following examples wrong:
{error_string}

Based on these errors, the problems with this prompt and the reasons are:
{error_feedback}

There is a list of former prompts including the current prompt, and each prompt is modified from its former prompts:
{trajectory_prompts}

Based on the above information, please write {steps_per_gradient} new prompts following these guidelines:
1. The new prompts should solve the current prompt's problems.
2. The new prompts should consider the list of prompts and evolve based on the current prompt.
3. Each new prompt should be wrapped with <START>and <END>.

The new prompts are: |

# D   TASK INPUT EXAMPLES

In this section, we show some input examples in several tasks for the base model. Specifically, our tasks fall into three categories: multi-choice selection, name entity recognition, and direct answer matching. As representative examples, we select *Penguins in A Table*, *NCBI*, and *Subjective* to illustrate the input format.

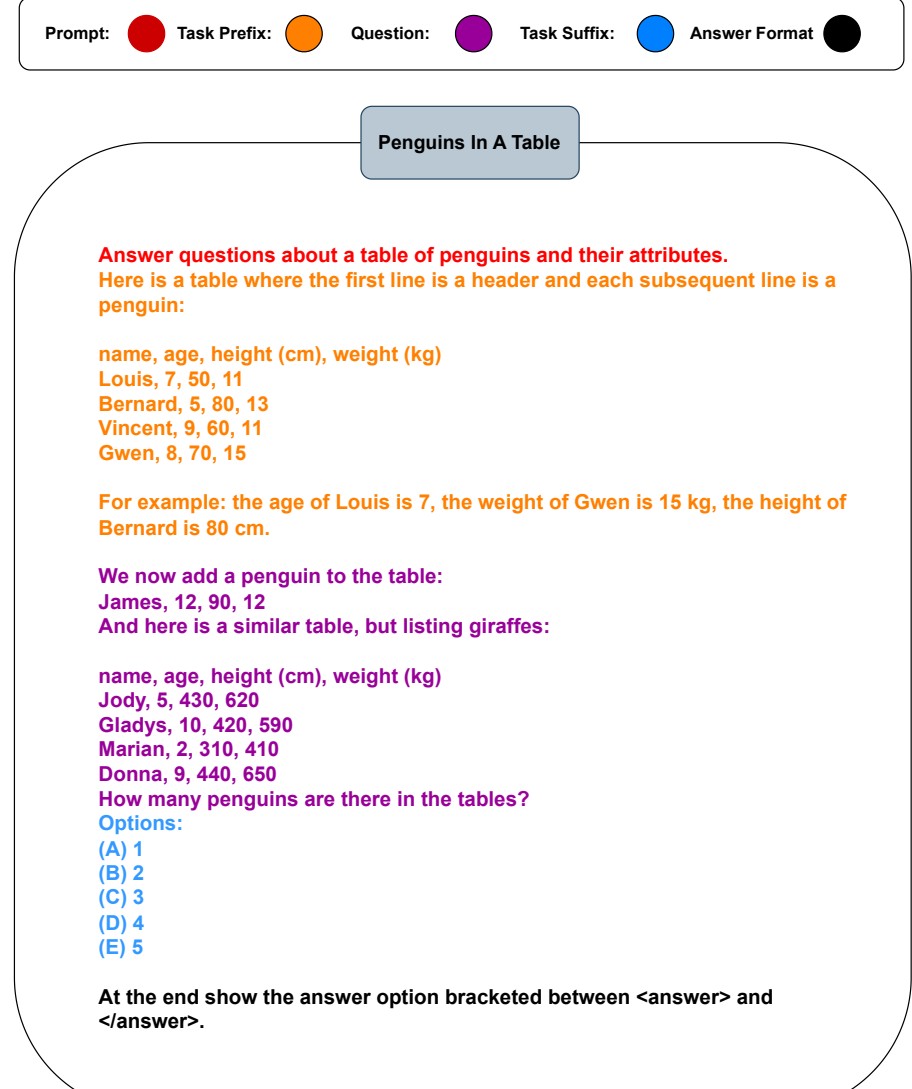

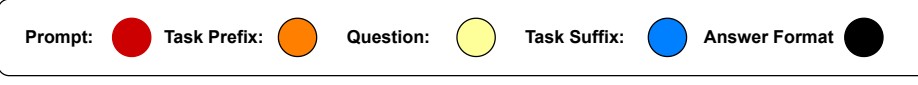

**NCBI**

**Extract the disease or condition from the sentence, if any is mentioned.**
**['Our', 'results', 'support', 'linkage', 'of', 'vWS', 'within', 'a', 'region', 'of', 'tightly',**
**'linked', 'markers', 'and', 'do', 'not', 'favour', 'locus', 'heterogeneity', 'of', 'the',**
**'disease', 'trait', '.']**
**Output the answer in this format:{entity_1,entity_2,....}. If no disease entities are**
**present, please output an empty list in this format: {}.**

**Subjective**

**Please perform Subjectivity Classification task. Given the sentence, assign a label**
**from ['subjective','objective']. Return label only without any other text.**
**Text: `` dreamcatcher `` tells the story of four young friends who perform a heroic**
**act - and are changed forever by the uncanny powers they gain in return .**
**Is the preceding text objective or subjective?**
**Options:**
**- Objective**
**- Subjective**

# E    CONVERGENCE OBSERVATION DETAILS

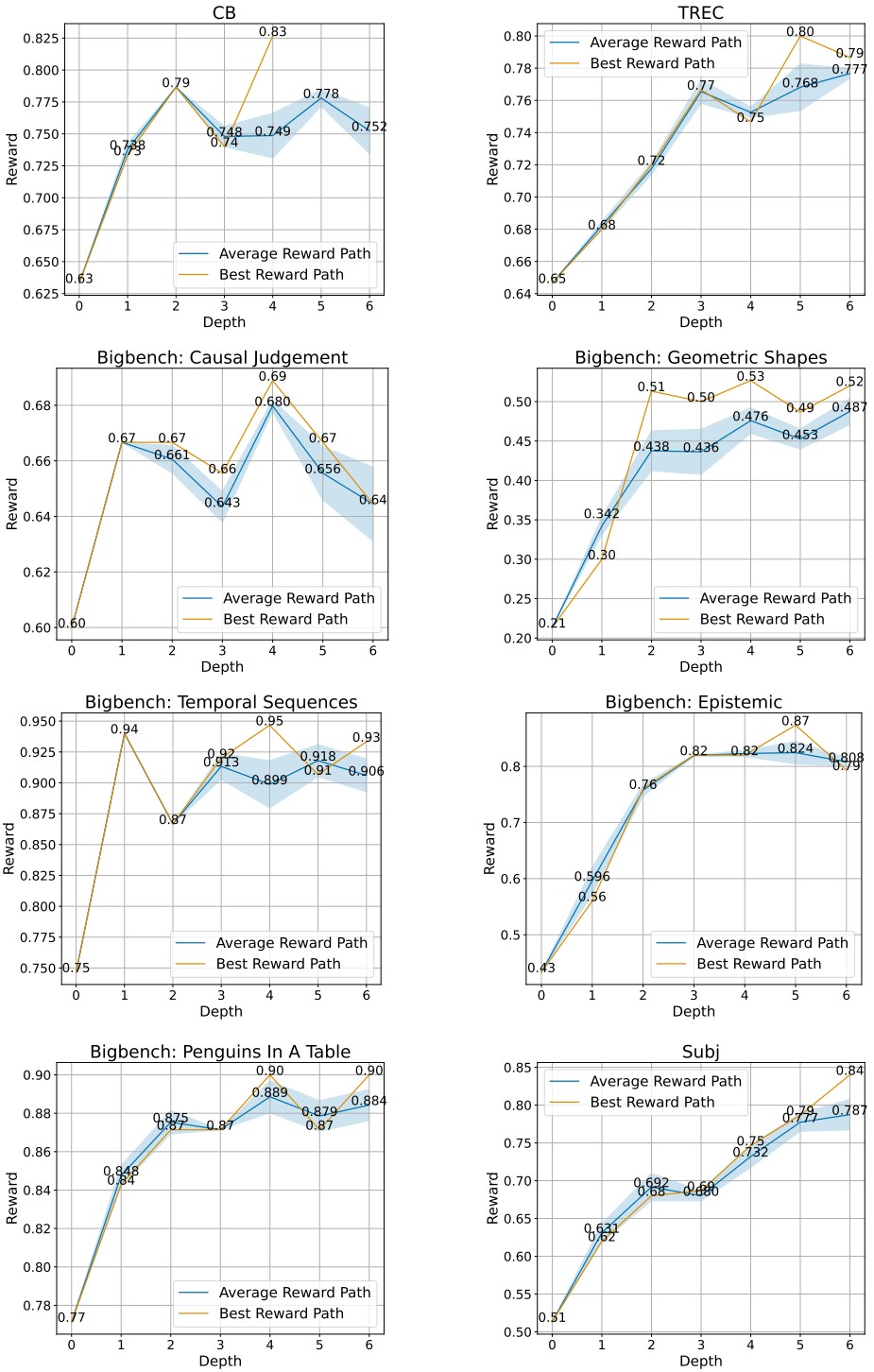

Figure 6: Convergence plots with the "Wide" setting. *expand_width* = 3, *num_samples* = 2, and *depth_limit* = 6. The Average Reward Path is the average reward of paths, and the blue area is the variance. The Best Reward Path is the path with the highest average reward, where the best node is selected as the node with the highest reward on the Best Reward Path.

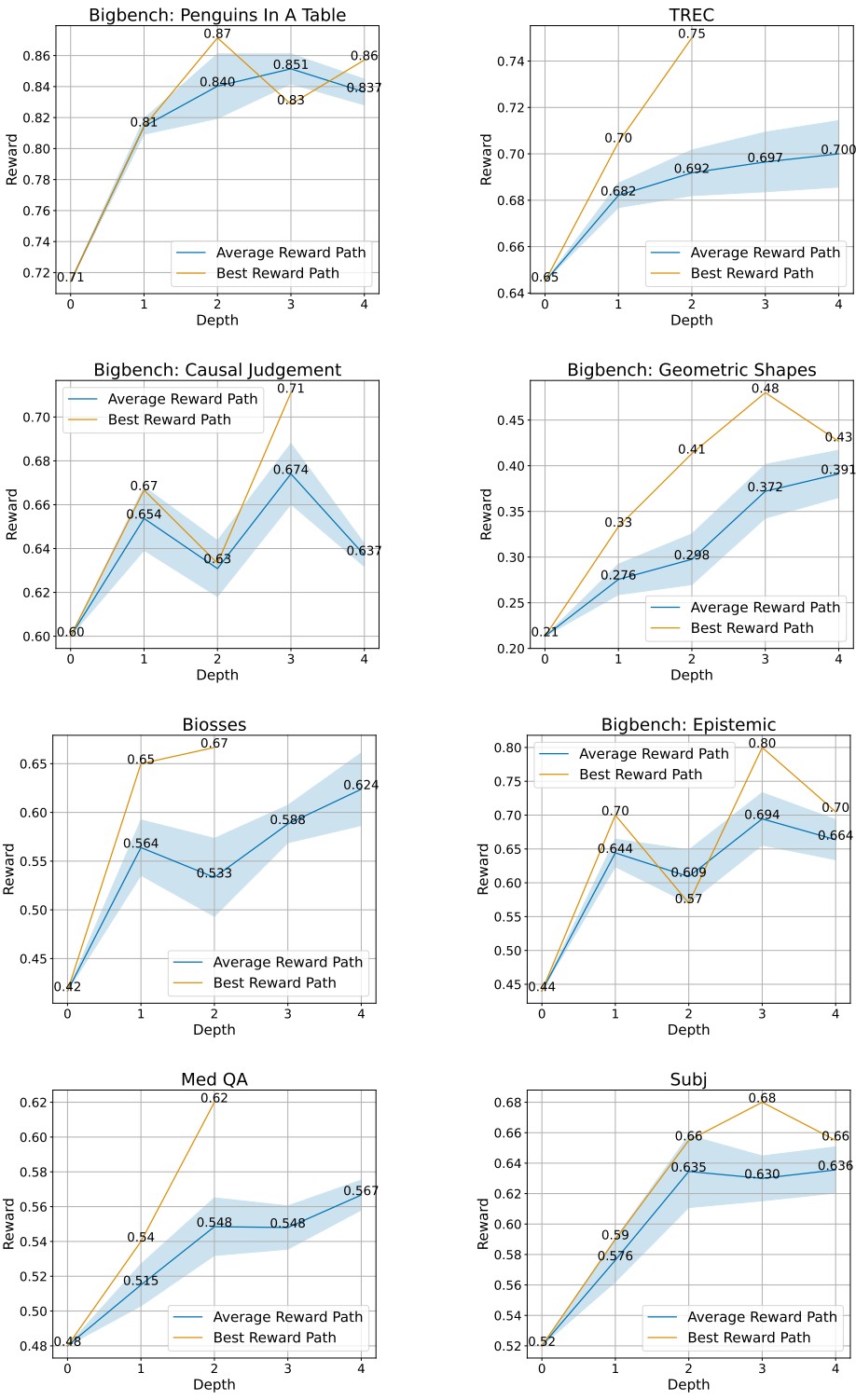

Figure 7: Convergence plots with the "Lite" setting. *expand_width* = 3, *num_samples* = 1, and *depth_limit* = 4. The Average Reward Path is the average reward of paths, and the blue area is the variance. The Best Reward Path is the path with the highest average reward, where the best node is selected as the node with the highest reward on the Best Reward Path.

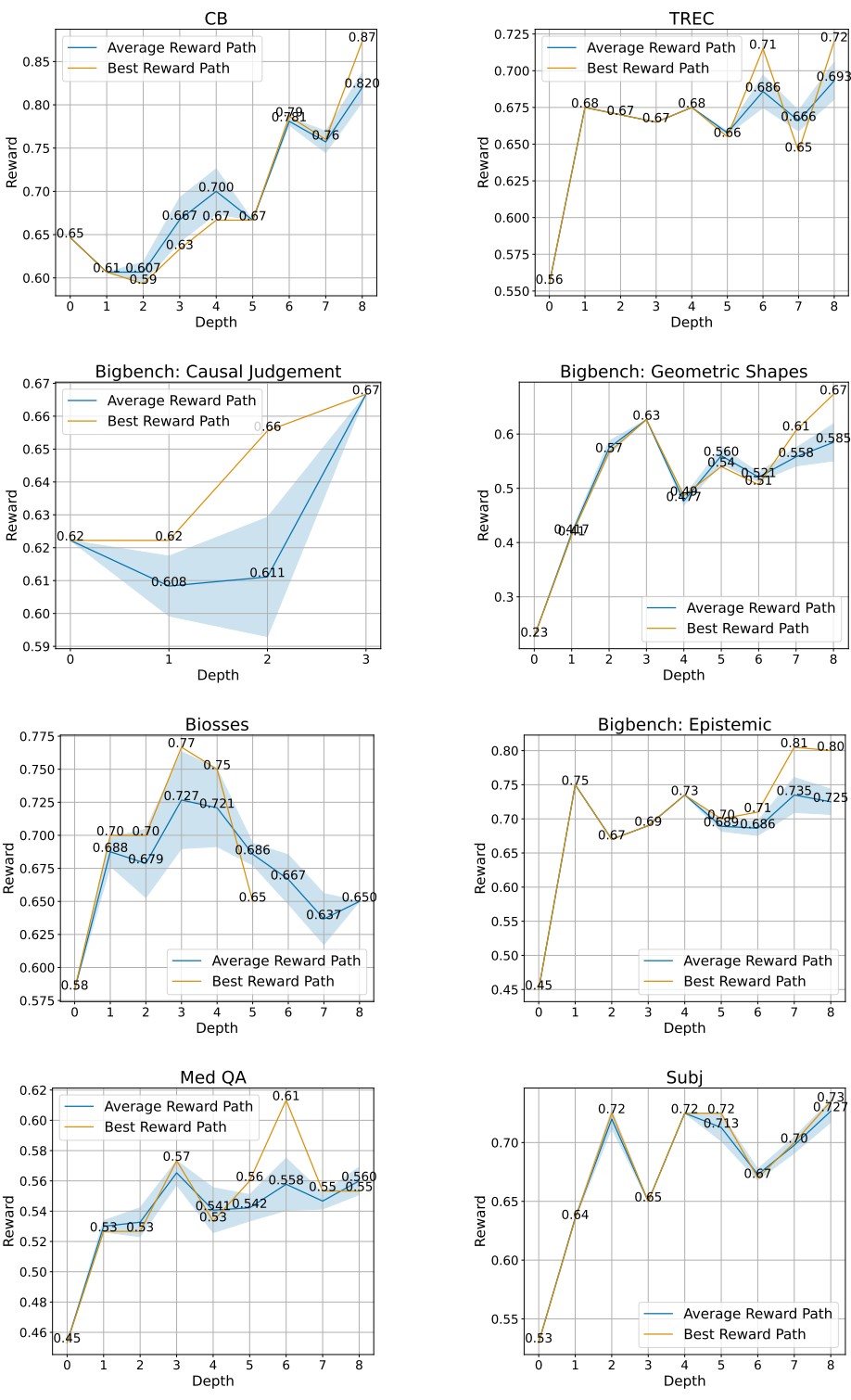

Figure 8: Convergence plots with the "Standard" setting. *expand_width* = 3, *num_samples* = 1, and *depth_limit* = 8. The Average Reward Path is the average reward of paths, and the blue area is the variance. The Best Reward Path is the path with the highest average reward, where the best node is selected as the node with the highest reward on the Best Reward Path.

## F    OPTIMIZED PROMPTS FROM PROMPTAGENT

Table 12: Prompt comparison for the Geometric Shapes task, including normal human prompt, APE-optimized prompt, and expert-level prompt optimized by PromptAgent. Both baselines mostly describe the task, while our expert prompt is composed of more complex structures and domain-specific insights, achieving superior performance. Bold text denotes **domain knowledge** usually handcrafted by domain specialists, but here automatically discovered by PromptAgent. We highlight different aspects of expert prompt with colors, including Task Description, Term Clarification, Solution Guidance, Exception Handling, Priority & Emphasis, Formatting. (Best view with colors)

| Approach | Optimized Prompt | Acc. |
|---|---|---|
| Human | Name geometric shapes from their SVG paths. | 0.227 |
| APE | "Determine the shape each SVG path element is drawing, then pair it with the corresponding letter from the available choices. In this case, C symbolizes hexagon, G is for pentagon, I signifies sector, and B stands for heptagon." | 0.490 |
| PromptAgent | In this task, you are tasked with interpreting SVG paths to determine the geometric figure they represent. The paths are delineated by commands: **'M' (move to), 'L' (line to), and 'A' (arc). An 'M' command initiates a path, potentially fragmenting a path into sub-paths, but it's crucial to not immediately view each 'M' as the starting point of a disconnected figure; often, they may continue the same geometric shape, manifesting as different sections within it. 'L' commands constitute line segments thus forming the boundaries of the figure. 'A' commands generate arcs, and depending on their sequence, can shape circles, sectors, elliptical figures, or other geometrical shapes through a continuous line of action. Note that an 'A' command followed by an 'L' could lead to specific shapes like sectors.** Examine the sequence and interplay of 'M', 'L', and 'A' commands, as they together mold the final geometric figure and significantly govern its continuity. Potential shapes to be identified can range from simple lines to complex polygons. 'None of the above' is only a valid response if otherwise stated in the task. As you formulate your answer, substantiate it with a clear explanation that encompasses the functionality of each command, their collective effect, sequence, and their correlational aspects. **In scenarios with multiple 'M' commands, refrain from arbitrarily breaking up the shape into disconnected figures; instead, visualize them contributing to different sections of the same shape.** Accurately count 'L' commands as they define the figure's sides, even when an 'M' command is present. For figuring out the entire geometric shape, meticulously examine all its components and commands, keeping an unbroken perception of the shape's progression, especially with multiple 'M' commands. Before finalizing your answer, recount the sides and arcs accurately - such a double-check ensures flawless identification of the geometric figure. | 0.670 |

Table 13: Prompt comparison for the Penguins In A Table task, including normal human prompt, APE-optimized prompt, and expert-level prompt optimized by PromptAgent. Both baselines mostly describe the task, while our expert prompt is composed of more complex structures and domain-specific insights, achieving superior performance. Bold text denotes **domain knowledge** usually handcrafted by domain specialists, but here automatically discovered by PromptAgent. We highlight different aspects of expert prompt with colors, including Task Description, Term Clarification, Solution Guidance, Exception Handling, Priority & Emphasis, Formatting. (Best view with colors)

| Approach | Optimized Prompt | Acc. |
|---|---|---|
| Human | Answer questions about a table of penguins and their attributes. | 0.595 |
| APE | Carefully scrutinize the provided table or tables. Understand the query in relation to the information given. Pinpoint the pertinent data and carry out the vital computations or comparisons to determine the right answer from the given choices. | 0.747 |
| PromptAgent | As you delve into a dataset of penguins, assess **essential attributes like names, ages, and gender**. Decode the significance of each attribute in the context of every penguin while **keeping in mind that the dataset may be modified, including addition or removal of penguins**. When such modifications are made, immediately revise your understanding, redo your computations, and ensure that your subsequent calculations consider these changes. The crux of your task is to identify relationships and patterns within the attributes, **giving special attention to the names and ages of the penguins**. For complex tasks, break them down into manageable chunks ensuring no essential detail is missed. When a change is made to the dataset, recompute your values taking into consideration these changes, paying extra attention to cumulative computations. **Ensure that your understanding of 'more than', 'less than', and 'equal to' is precise and that you correctly interpret these in context of the question.** Put into place a verification mechanism to authenticate the accuracy of your solutions, stating out your understanding of the query and the assumptions you have made to resolve it. **Bear in mind that tasks may require you to combine the dataset with additional external information, this may include understanding age disparities outside explicit lifespan parameters, identifying common names linked to gender, or recognizing names associated with famous individuals.** Document your matters of interest meticulously and maintain rigorous accuracy levels in your calculations to prevent errors. Stay nimble-footed in reshaping your analytical approach based on each new query. This might include uncovering numerical patterns, comprehending inherent data natures, or liaising with external sources for a more thorough understanding. **Most importantly, prior to making a comparison within attributes such as age or height, conduct a thorough investigation of all values under that attribute.** Understand the premise of each question before springing to deductions, and remember, any change in the dataset denotes a new starting point for the following computational steps to maintain accuracy. | 0.873 |

Table 14: Prompt comparison for the Epistemic Reasoning task, including normal human prompt, APE-optimized prompt, and expert-level prompt optimized by PromptAgent. Both baselines mostly describe the task, while our expert prompt is composed of more complex structures and domain-specific insights, achieving superior performance. Bold text denotes **domain knowledge** usually handcrafted by domain specialists, but here automatically discovered by PromptAgent. We highlight different aspects of expert prompt with colors, including Task Description, Term Clarification, Solution Guidance, Exception Handling, Priority & Emphasis, Formatting. (Best view with colors)

| Approach | Optimized Prompt | Acc. |
|---|---|---|
| Human | Determine whether one sentence entails the next. | 0.452 |
| APE | Determine whether the hypothesis is directly implied by the premise or not. If the premise's statement is a direct claim or conviction of the individual mentioned in the hypothesis, choose 'entailment'. However, if the premise is formed on the belief or supposition of someone other than the subject in the hypothesis, opt for 'non-entailment'. | 0.708 |
| PromptAgent | Your task is to critically analyse the primary sentence, known as the 'premise', with the objective of determining whether it unequivocally supports the truth value of the subsequent sentence or 'hypothesis'. The relationship between the premise and hypothesis can be classified as 'Entailment' or 'Non-Entailment'. Label it as 'Entailment' if the premise provides **robust evidence substantiating the truth of the hypothesis without requiring additional context**. If, however, the corroboration of the hypothesis by the premise is not entirely explicit, select 'Non-Entailment'.

Deciphering the semantics within the sentences is crucial for your final decision. **Terms such as 'assumes', 'believes', 'thinks', 'feels', 'suspects', and their likes should be respected for their capacity to introduce uncertainty and subjectivity, and not perceived as conclusive proof of the hypothesis, regardless of whether they form part of nested beliefs or not. Also, a detailed premise does not necessarily negate a more generalized hypothesis. For example, a premise that mentions a 'full face mask' correlates to a hypothesis that states a 'mask'.**

During your evaluation, maintain a keen focus on factual and logical reasoning, always bearing in mind that personal beliefs or experiences should be incorporated into your review only if they are inherently connected to the factual content of the statements. **However, these should be understood as subjective truths in the context of the individual's perspective and should not be taken as objectively verifiable truths.** Upon deciding between 'Entailment' or 'Non-Entailment', articulate your explanations in a concise manner, warranting that you desist from making precipitous conclusions or unsupported assumptions. Your judgement should be firmly anchored in the logical and factual ties existing within the premise and hypothesis, renouncing any incidental inferences or personal interpretations.
**Exercise restraint in passing verdicts on the truth value or validity of personal beliefs, unless they have a direct bearing on the factual correlation between the premise and the hypothesis**. During your estimation, mindfully weigh the extent of uncertainty introduced by expressions of belief or suspicion against the imperative for factual precision when establishing the entailment. | 0.806 |

Table 15: Prompt comparison for the Object Counting task, including normal human prompt, APE-optimized prompt, and expert-level prompt optimized by PromptAgent. Both baselines mostly describe the task, while our expert prompt is composed of more complex structures and domain-specific insights, achieving superior performance. Bold text denotes **domain knowledge** usually handcrafted by domain specialists, but here automatically discovered by PromptAgent. We highlight different aspects of expert prompt with colors, including Task Description, Term Clarification, Solution Guidance, Exception Handling, Priority & Emphasis, Formatting. (Best view with colors)

| Approach | Optimized Prompt | Acc. |
|---|---|---|
| Human | Count the overall number of all items. | 0.612 |
| APE | Calculate the overall total of all items even those spoken in groups. | 0.716 |
| PromptAgent | Carefully analyze the given information. Catalog each item mentioned and denote any explicitly defined quantities. **If an item - quantity is not stated, assume it as a single unit. However, for an item with a specified quantity, make sure to count each unit separately and include it in your total count.** If collective terms or categories are identified, break them down into their individual components and reasonably associate each with its stated count. Proceed to calculate a comprehensive total for such categories ensuring the sum includes all individual units, not the number of subsets or types. **Remember that each item has its unique count, but items related or falling under a common category should be tabulated as such, with their individual quantities precisely contributing to the final count.** Avoid making assumptions about the nature or categorization of items and adhere to commonly accepted definitions and classifications. Review your work to ensure accuracy and to avoid mistakes in counting. **Modify your strategy if required by considering items within varying categories, types, or subtypes**. Eventually, summarize the count indicating the specific quantity for each identified item or category and a total count of units, not categories, or provide a comprehensive overview as explicitly requested. | 0.86 |

Table 16: Prompt comparison for the Temporal Sequences task, including normal human prompt, APE-optimized prompt, and expert-level prompt optimized by PromptAgent. Both baselines mostly describe the task, while our expert prompt is composed of more complex structures and domain-specific insights, achieving superior performance. Bold text denotes **domain knowledge** usually handcrafted by domain specialists, but here automatically discovered by PromptAgent. We highlight different aspects of expert prompt with colors, including Task Description, Term Clarification, Solution Guidance, Exception Handling, Priority & Emphasis, Formatting. (Best view with colors)

| Approach | Optimized Prompt | Acc. |
|---|---|---|
| Human | Answer questions about which times certain events could have occurred. | 0.72 |
| APE | Identify the period when the individual was unnoticed and had the possibility to visit the specified place before its closing time. | 0.856 |
| PromptAgent | By examining the series of daily activities of an individual, pinpoint when they were free and when they were busy. Use these open slots to dictate when they could possibly engage in other activities. **Upon waking up, a person does not instantly become occupied. Take into account any potential restrictions or closed times and use these as an indicator that the event cannot take place during these hours. An overlap of activities is unallowable, so ensure there is no overlap while creating a timeline.** Cross-check the free time slots with the functioning hours of the potential event to accurately derive the most likely time interval for the event to take place. | 0.934 |

Table 17: Prompt comparison for the Causal Judgment task, including normal human prompt, APE-optimized prompt, and expert-level prompt optimized by PromptAgent. Both baselines mostly describe the task, while our expert prompt is composed of more complex structures and domain-specific insights, achieving superior performance. Bold text denotes **domain knowledge** usually handcrafted by domain specialists, but here automatically discovered by PromptAgent. We highlight different aspects of expert prompt with colors, including Task Description, Term Clarification, Solution Guidance, Exception Handling, Priority & Emphasis, Formatting. (Best view with colors)

| Approach | Optimized Prompt | Acc. |
|---|---|---|
| Human | Answer questions about causal attribution. | 0.47 |
| APE | "For each situation, decide if the result was caused deliberately or not. If the individual or party behind the event was aware of the potential result and chose to go ahead, select 'A'. If they didn't intend the result to happen, even if they knew it could possibly occur, select 'B'." | 0.57 |
| PromptAgent | Respond to inquiries about causal attribution, focusing on the entity or entities specifically highlighted in the question. **Carefully investigate multi-factorial causes that may operate simultaneously and independently**, and discern the underlying intentions behind an individual's actions. Differentiate between immediate and incidental origins and identify the contribution of each factor in creating the outcome. **Examine the interplay of causes within the immediate situation and larger systemic frameworks**. Maintain uncompromising adherence to the details provided within the context and restrain from making assumptions unsupported by the evidence presented. **Always consider the complexity of multiple causes contributing to a single effect and resist attributing the effect to a singular cause. Recognize the possibility of synergy amongst causes and its resultant effects**. | 0.67 |

Table 18: Prompt comparison for the NCBI task, including normal human prompt, APE-optimized prompt, and expert-level prompt optimized by PromptAgent. Both baselines mostly describe the task, while our expert prompt is composed of more complex structures and domain-specific insights, achieving superior performance. Bold text denotes **domain knowledge** usually handcrafted by domain specialists, but here automatically discovered by PromptAgent. We highlight different aspects of expert prompt with colors, including Task Description, Term Clarification, Solution Guidance, Exception Handling, Priority & Emphasis, Formatting. (Best view with colors)

| Approach | Optimized Prompt | F1 score. |
|---|---|---|
| Human | Extract the disease or condition from the sentence, if any is mentioned. | 0.521 |
| APE | If any disease or condition is mentioned in the sentence, extract it. | 0.576 |
| PromptAgent | You're tasked with extracting diseases or conditions from the given sentence, remember to be cautious and **avoid incorporating any associated elements such as inheritance patterns (like autosomal dominant), genes or gene loci (like PAH), proteins, or biological pathways**. The task does not entail making assumptions or inferences about the disease names based on other advanced biological terms in the context. **Consider both specific diseases and broader categories, and remember diseases and conditions can also appear as common abbreviations or variations**. Provide the identified diseases or conditions in this format: {entity_1,entity_2,....}. If there are no diseases or conditions present, output an empty list in this form: {}. **Note that the term 'locus' should be recognized as a genomic location and not a disease name**. | 0.645 |

Table 19: Prompt comparison for the Biosses task, including normal human prompt, APE-optimized prompt, and expert-level prompt optimized by PromptAgent. Both baselines mostly describe the task, while our expert prompt is composed of more complex structures and domain-specific insights, achieving superior performance. Bold text denotes **domain knowledge** usually handcrafted by domain specialists, but here automatically discovered by PromptAgent. We highlight different aspects of expert prompt with colors, including Task Description, Term Clarification, Solution Guidance, Exception Handling, Priority & Emphasis, Formatting. (Best view with colors)

| Approach | Optimized Prompt | Acc. |
|---|---|---|
| Human | This is a biomedical sentence similarity task. Please carefully read the following sentences and rate the similarity of two input sentences. Choose between 'not similar', 'somewhat similar' and 'similar' | 0.55 |
| APE | "Examine the two given sentences and assess their content similarity. Choice A (not similar) should be selected if the sentences discuss entirely different topics or concepts. Choose option B (somewhat similar) if they have some common points but also contain differences. Select option C (similar) if the sentences primarily convey the same message or could be used in place of one another." | 0.7 |
| PromptAgent | For this task, you are asked to perform a biomedical sentence similarity evaluation. Examine the two input sentences and evaluate their similarity, not only taking into account common terms or concepts **but also the complex scientific language, specific processes, and unique subject matter they delve into**. **Consider not only the subject matter but also the intended purpose like whether they both describe a process, report a finding, or detail a method or technique**. Rate the similarity as 'not similar' if their subject matter or emphasis is distinct, 'somewhat similar' if they discuss related topics or share some details but are not entirely identical, and 'similar' if the sentences precisely mirror each other in topic and conclusions. Remember, this task requires more than a cursory scan of keywords - focus on the nuanced meanings, pay attention to the degree at which the discussed concepts or processes are general or specific, and strive for a comprehensive understanding of the contents. | 0.75 |

Table 20: Prompt comparison for the Med_QA task, including normal human prompt, APE-optimized prompt, and expert-level prompt optimized by PromptAgent. Both baselines mostly describe the task, while our expert prompt is composed of more complex structures and domain-specific insights, achieving superior performance. Bold text denotes **domain knowledge** usually handcrafted by domain specialists, but here automatically discovered by PromptAgent. We highlight different aspects of expert prompt with colors, including Task Description, Term Clarification, Solution Guidance, Exception Handling, Priority & Emphasis, Formatting. (Best view with colors)

| Approach | Optimized Prompt | Acc. |
|---|---|---|
| Human | Please use your domain knowledge in medical area to solve the questions. | 0.508 |
| APE | "For every presented clinical situation, scrutinize the symptoms and specifics given. From the options A-E, choose the one that best pinpoints the cause or diagnosis of the stated condition." | 0.47 |
| PromptAgent | Leveraging particularly your comprehensive medical expertise, handle each presented scenario as you would a complicated puzzle requiring careful, unbiased assessment. **Each nugget of information - from patient age, gender, lifestyle, symptoms, lab results, and past medical history**, to recent activities that may be relevant to their condition, plays an equally important role in shaping your judgement. **Becoming cognizant of the fact that medical conditions can manifest uniquely in different individuals is crucial; avoid precipitating conclusions merely on the basis of stereotypical symptoms**. Instead, employ a deep understanding of the variety of medical conditions to critically evaluate each symptom's relevance, ensuring that undue bias is not allocated to particular symptoms over others. Particularly, pay attention to common symptoms over rare ones unless otherwise indicated. Break down assumptions and consider the most likely cause in a given context. **Do not overlook the importance of demographic details and their correlation with symptoms, especially when a symptom hints at a particular physiological state, like menopause**. Through meticulous examination, ensure you grasp the nuances in each query's context, **with keen focus on the developmental stages in children and the specific challenges they entail**. Capture the timelines of symptoms, understanding that often, a diagnosis relies significantly on the onset and duration of these symptoms. Once conclusions begin taking shape, undertake an exhaustive cross-verification exercise with the available multiple choice answers. Evaluate these options for relevance and decide their probability on the specifics of the given case. Abstain from dismissing potential answers at first glance, but rather advocate for an intensive assessment of all. Approach scenarios similar to solving a complex jigsaw puzzle. **Each distinct symptom, lab result, past medical history, and timing forms an integral component that lends weight to a deeper comprehension of the patient's present condition**. The endgame extends beyond merely achieving precision and a comprehensive enquiry but ensures that your conclusions do not yield overgeneralization or oversimplification towards the diagnosis and treatment therein. **Examine closely every symptom in relation to the disease and differentiate those that are side effects of treatment. Be cautious when multiple symptoms present simultaneously, to avoid confusion**. The imprint of your insight should reflect a holistic understanding of the case, zooming into the most probable diagnosis or treatment strategy that suits the breadth of data at disposal. | 0.57 |

Table 21: Prompt comparison for the Subjective task, including normal human prompt, APE-optimized prompt, and expert-level prompt optimized by PromptAgent. Both baselines mostly describe the task, while our expert prompt is composed of more complex structures and domain-specific insights, achieving superior performance. Bold text denotes **domain knowledge** usually handcrafted by domain specialists, but here automatically discovered by PromptAgent. We highlight different aspects of expert prompt with colors, including Task Description, Term Clarification, Solution Guidance, Exception Handling, Priority & Emphasis, Formatting. (Best view with colors)

| Approach | Optimized Prompt | Acc. |
|---|---|---|
| Human | Given the text, choose between 'subjective' and 'objective'. | 0.517 |
| APE | Determine whether the provided text is stating facts and details (Objective) or expressing personal views, emotions, or choices (Subjective). | 0.696 |
| PromptAgent | Examine the given text and decide whether it is 'subjective' or 'objective'. Define the narrative as 'subjective' **if it seems to be significantly swayed by the author's personal emotions, viewpoints, or beliefs**. Conversely, 'objective' narratives should **impartially depict facts or scenarios, devoid of personal prejudices, preconceived beliefs, and the author's own convictions**. **It is essential to understand that emotionally-dense language, vivid descriptions or depiction of characters' emotional states do not always hint at subjectivity**. They may just serve to represent situations authentically without conveying the author's personal standpoint. **Unconventional punctuation, dialogues or queries do not inherently contribute to authorial subjectivity**. Draw a clear distinction between the author's and characters' subjectivity; misinterpreting a character's subjectivity as the author's personal bias is a common pitfall. The priority is to extract the author's tendency within the narrative, rather than focusing on the characters. Utilize these directives to critically analyze the text. | 0.806 |

Table 22: Prompt comparison for the TREC task, including normal human prompt, APE-optimized prompt, and expert-level prompt optimized by PromptAgent. Both baselines mostly describe the task, while our expert prompt is composed of more complex structures and domain-specific insights, achieving superior performance. Bold text denotes **domain knowledge** usually handcrafted by domain specialists, but here automatically discovered by PromptAgent. We highlight different aspects of expert prompt with colors, including Task Description, Term Clarification, Solution Guidance, Exception Handling, Priority & Emphasis, Formatting. (Best view with colors)

| Approach | Optimized Prompt | Acc. |
|---|---|---|
| Human | Tag the text according to the primary topic of the question. Choose from (A) Abbreviation, (B) Entity, (C) Description and abstract concept, (D) Human being, (E) Location, (F) Numeric value | 0.742 |
| APE | "Tag the text according to the primary topic of the question. Select 'Human being' (D) if the question **revolves around a person**. Opt for 'Description and abstract concept' (C) if the question **requires an explanation or description of a concept**. Choose 'Location' (E) if the question is about a specific place. If the question refers to a particular object or thing, then select 'Entity' (B). If the question **involves data or a length of time**, opt for 'Numeric value' (F). Disregard 'Abbreviation' (A) since it's not related to any of the questions." | 0.834 |
| PromptAgent | For the question given above, determine the type of response it is aiming to elicit, then assign the most fitting label from the following: (A) Abbreviation, (B) Tangible and Intangible Entity **(including distinct terms, theories, inventions, phenomena)**, (C) Description and Abstract Concept **(concerning explanations, clarifications, theoretical ideas)**, (D) Individual and Collective Humans **(encompassing distinct persons, the creators of certain works, groups, organizations)**, (E) Location, or (F) Numeric Value **(containing numeric figures, dates, timings, quantities)**. The key is the answer-type the question is seeking, not other elements in the question. Your assigned label should prioritize the primary response over additional details. If a solo label does not closely address the entire answer intent of the question, then you may assign more than one. The label should reflect the assumed answer's nature, not the mere question's content or incidental features. Place the label you consider most fitting for the question's main intention. | 0.886 |

Table 23: Prompt comparison for the CB task, including normal human prompt, APE-optimized prompt, and expert-level prompt optimized by PromptAgent. Both baselines mostly describe the task, while our expert prompt is composed of more complex structures and domain-specific insights, achieving superior performance. Bold text denotes **domain knowledge** usually handcrafted by domain specialists, but here automatically discovered by PromptAgent. We highlight different aspects of expert prompt with colors, including Task Description, Term Clarification, Solution Guidance, Exception Handling, Priority & Emphasis, Formatting. (Best view with colors)

| Approach | Optimized Prompt | Acc. |
|---|---|---|
| Human | Read carefully the following premise and hypothesis, and determine the relationship between them. Choose from 'contradiction', 'neutral' and 'entailment'. | 0.714 |
| APE | "Ascertain the link between the premise and the hypothesis. If the hypothesis happens to be a rational outcome or inference from the premise, label it as an 'entailment'. If the hypothesis presents a contrasting scenario or clashes with the premise, categorize it as a 'contradiction'. In case the hypothesis neither disputes nor is it derived from the premise, term it as 'neutral'." | 0.8036 |
| PromptAgent | Your task is to delve deeply into the provided premise and hypothesis. Highlight explicit, central information and important entities mentioned in the dialogue while considering multiple ways the same thought could be delivered through language. **Acknowledge that a hypothesis might reflect, rephrase, or reiterate ideas from the premise, possibly in a simplified manner.** However, remember that **mere verbatim repetition does not automatically signal 'entailment'.** The reiteration in the hypothesis should represent a pivotal idea in the premise for it to be categorized as entailment. **If the hypothesis asserts something diametrically opposed to what's stated in the premise, mark it as a 'contradiction'. Reserve 'neutral' for scenarios where the premise and the hypothesis appear disconnected or do not exhibit any clear relationship.** Be vigilant while dealing with ambiguities, and strive to decode them in the context of the hypothesis. Do not allow nuanced or hypothetical statements distract from identifying the primary idea in the hypothesis. Know that your classifications, 'entailment', 'contradiction', or 'neutral', should mirror the essential relationship derived strictly from the premise and the hypothesis, without the influence of personal opinions or conclusions. Prioritize understanding the core intention and context of the conversation over mere repetition of words or phrases. | 0.911 |

