# OpenReview forum: "PromptAgent: Strategic Planning with Language Models Enables Expert-level Prompt Optimization"
_ICLR.cc/2024/Conference — ICLR 2024 poster_

### Official Review · Reviewer_j86n · 2023-10-28

**Soundness:** 3 good
**Presentation:** 3 good
**Contribution:** 2 fair
**Rating:** 6
**Confidence:** 4

**Summary:**

The paper proposes a strategic prompt engineering method by utilizing Monte Carlo Tree Search where a base model collects errors from the training data set and an optimizer model provides feedback based on the collected errors to further optimize the prompt. Once a new prompt is composed, the hold-out set will be used to gauge its performance based on a given reward function. The method is evaluated on 12 curated tasks from three domains where results show its superior performance compared to a few alternative approaches including human curated prompt, Chain of Thoughts (CoT), and a couple of optimization methods including GPT agent and Automatic Prompt Engineering. Authors study the the effect of strategic planning aspect of their method through a set of ablation studies by considering various alternatives through single Monte Carlo Search, greedy, and Beam search where results show the benefit of considering exploitation vs. exploration trade offs in exploring the search space. Authors study prompt generalization and exploration efficiency as well.

**Strengths:**

- The paper is well written and easy to follow.
- Related literature is covered.
- Authors have conducted comprehensive experiments to evaluate the performance of their proposed method against a set of alternative methods and ablation studies shed further lights on the effectiveness of the proposed strategic exploration of the search space using MCTS.
- Appendix covers useful details about the hyper parameters and the evaluation data sets and details of the studied methods.

**Weaknesses:**

- **Novelty** of the proposed method is arguable. Use of MCTS in prompt engineering is not novel. However, the way authors have incorporated a base and an optimizer model in their implementation and the prompts used to incorporate the error feedback into the existing state (+ the empirical studies) can be considered as the main contributions of this work.
- **Contribution**: Based on the details provided in the Appendix section, it can be seen that the amount of details covered in prompts generated by APE is not comparable with that of the proposed method. The prompts generated by the proposed method are significantly lengthier and cover more details compared which is hard to justify. APE supposedly uses Monte Carlo Search to iteratively propose and select prompts, therefore, it is expected to see more details getting added to the original prompt over each iteration which is not reflected in the samples that I see in the Appendix section. **This makes me conclude that the additional gain from the proposed method by authors come from the way they instruct the optimizer method to incorporate the feedback into the existing state rather than utilization of MCTS** which is the main claim of the paper.
However,
- Clarity [minor]: It's ok that authors have covered the details of the selection, expansion, simulation, and back-propagation in MCTS and it can be very useful for general audience with less context, however, I was hoping to read more details about how authors have implemented the expansion stage. Appendexi briefly touches base under "Implementation details" sub-section and mentions "We sample 3 batches, and for each batch, we generate multiple new prompts". It would be good if authors can further explain how they generate new prompts for each batch.

**Questions:**

- Page 2: What does it mean when you say "bad at managing multiple errors"
- It's not clear how the authors have picked the hyper parameters including number of iterations and exploration vs. exploitation related parameter.

Minor:
p 1: prompting engineering => prompt engineering
P 3: without as less as => with as less as

**Details Of Ethics Concerns:**

While I do not see any ethical issues with this paper in particular, I want to highlight an ethical risks that is common across all machine learning methods that could be more concerning for prompt engineering tasks since we are dealing with human readable sentences rather than a hard to understand model:
One should proceed with caution incorporating the optimizer LLM feedback into the prompt. The underlying optimizer LLM may summarize the error feedback by incorporating a biased/discriminatory statement into the existing prompt and there is no supervision/mechanism in the proposed method to address for such scenarios. e.g. consider the task of loan approval based on some information about an individual and imagine based on an error for an individual from a certain demographic, the optimizer decides to explicitly add an statement about a certain race/demographic or any other protected attributes in the prompt.

---

> ### Author Response · Authors · 2023-11-22
> **Response to Reviewer j86n (Part 1)**
>
> We thank Reviewer j86n for the detailed evaluation of our work, for acknowledging that our soundness is good, our paper is well-written and easy to follow, related literature is covered, our experiments are comprehensive, shedding further light on the effectiveness of the proposed strategic exploration of the search space using MCTS, and our appendix provides useful details. We find reviewer j86n has a slightly different understanding of the core novelty and contribution of our work (we though thank the reviewer for the self-debating). We'll thoroughly address and clarify these mismatched understandings as follows, hoping the reviewer will be on the same page with us at the end of this discussion.
>
> (a) **Clarifying the novelty.** We kindly redirect the reviewer to our global response (a), where we thoroughly address the technical novelty and core contributions in PromptAgent. Regarding the concern from reviewer j86n, we argue that interpreting PromptAgent simply as "use of MCTS in prompt engineering" is neither accurate nor faithful to reflect its core contributions.
>
> However, we're glad to see reviewer j86n acknowledge our other novelties and contributions as base/optimizer formulation and incorporate error feedback. Given the partial consensus we've made here, we hope we could acknowledge this as the strength of PromptAgent rather than its weakness.
>
> (b) **Clarifying the contribution of MCTS.** When comparing APE with PromptAgent, reviewer j86n admits the importance of the error feedback mechanism, instead of the utilization of MCTS. The reason probably is that they find APE prompts in our paper (appendix F) are not detailed enough, making them question the usefulness of iterative search methods in deriving expert-level detailed prompts.
>
> 1. First, we agree with the reviewer on **the importance of error feedback** in inducing domain knowledge. We implemented two versions of the Monte Carlo search using the same error feedback mechanism: MC-single: iteratively generates one path with a depth of 8, and MC-depth=1: directly sample 72 prompts from the initial prompt for the selection. Both can be viewed as variants of APE within the PromptAgent framework. Note that the original APE doesn't have the error feedback mechanism and, thus, cannot effectively induce detailed domain knowledge and instructions into the prompt. [The figure from the original APE paper](https://drive.google.com/file/d/1PNcFqRcGAVH69ABUZ3ZvVx8xpN42zivk/view?usp=sharing) shows their optimized prompts are actually short with little domain knowledge.
>
> - We now show prompts from our APE and MC-dept=1, whose major difference is APE doesn't have the error feedback. The following examples also show that even with only a single sampling method, the error feedback optimization step can integrate much more domain knowledge than APE.
>
>
> MC-depth=1 prompt for the Subjective task:
> ```
> Analyze the presented text in its entirety. Do not focus on individual words or phrases, and remember that objective descriptions can also include factual information about emotional states or conditions. Choose whether the text is 'subjective', characterized usually by perspectives, opinions, beliefs, and feelings, or 'objective', typically including verifiable facts or descriptions, regardless of its tone, style, or phrasing. Keep in mind that the text's intention and the nature of its content have to be distinguished. The text can be informal and still convey objective information. With this in mind, classify the given text as either 'subjective' or 'objective'.
> ```
>
> APE prompt for the Subjective task:
> ```
> Determine whether the provided text is stating facts and details (Objective) or expressing personal views, emotions, or choices (Subjective).
> ```
>
> MC-depth=1 prompt for the Geometry task:
> ```
> Given an SVG path, interpret the formula to deduce the specific geometric shape it forms when all commands are completed. Keep in mind that the path commands work together to create a figure which represents a simple or complex geometric shape, and not just individual lines. Be sure to count vertices and pay attention to the structure created. Lastly, for this task, we are only interested in identifying geometric shapes such as circles, triangles, pentagons, hexagons, heptagons, kites, octagons, rectangles, sectors or lines. Try to match the resulting figure with one of these provided options.
> ```
>
> APE prompt for the Geometry task:
> ```
> Determine the shape each SVG path element is drawing, then pair it with the corresponding letter from the available choices. In this case, C symbolizes hexagon, G is for pentagon, I signifies sector, and B stands for heptagon.
> ```

---

> > ### Author Response · Authors · 2023-11-22
> > **Response to Reviewer j86n (Part 2)**
> >
> > 2. Second, regarding the importance of the utilization of MCTS, we were hoping **our ablation study on search variants** has shown enough evidence for the superiority of MCTS. By keeping the same state transition and action/error feedback generation, we ablate MCTS with many search variants, including Monte Carlo search, beam search, and greedy search. The reviewer j86n also acknowledges this in the strength section, with **"ablation studies shed further lights on the effectiveness of the proposed strategic exploration of the search space using MCTS."** We now present the new results with two MC variants and other search variants in the following table to further demonstrate the necessity of MCTS in PromptAgent.
> >
> > RTable 6: Thorough ablation study on search variants
> > | Category | MC-single| MC-depth=1 | Greedy | Beam  | MCTS  |
> > |----------|-------|-------|--------|-------|-------|
> > | Penguins | 0.759  |0.772| 0.823  | 0.810 | 0.873 |
> > | Biosses  | 0.542  |0.575| 0.675  | 0.700 | 0.750 |
> > | Geometry | 0.482  |0.490| 0.610  | 0.545 | 0.670 |
> > | Causal   | 0.603  |0.650| 0.610  | 0.660 | 0.670 |
> > | Subj     | 0.628  |0.692| 0.765  | 0.778 | 0.806 |
> > | Average  | 0.602  |0.635| 0.697  | 0.698 | 0.754 |
> >
> > RTable 6 presents experiments applying the same error feedback optimization step on all various search strategies. Note that MC-depth=1, Greedy, and Beam have the same exploration efficiency (72 nodes) and the same depth (8). Moreover, MC-depth=1 can be viewed as an error feedback-based APE method, which doesn't perform very well. We can see that although we have a good method to incorporate information from error feedbacks and generate new prompts, a good searching algorithm is a must for efficiently and effectively exploring the prompt space to generate a high-performance prompt.
> >
> > In summary, we are confident to conclude that both error feedback-based optimization steps and advanced strategic planning algorithms like MCTS are essential for the success of PromptAgent.
> >
> > (c\) **Clarifying a minor comment: more details on the expansion step.** We're happy to clarify more details of the expansion step. During the expansion step, "expand_width=3" batches of examples will be sampled from the training set. We will perform state transit on each of these batches, and each state transit will generate "steps_per_gradient" number of new prompts (the default number is 1, and it is 2 in the Width experiment setting), so basically, in each Expansion, we perform "expand_width=3" state transit and get "expand_width"$\times$"steps_per_gradient" number of new prompts. In each state transit, the batch will be fed to the base model to collect the errors. If there is no error, a new batch will be sampled until errors are found. The errors will be formatted using "error_string"  and inserted into "error_summarization" to get the error feedbacks from the optimizer. Then, the error_string, error feedback, and the trajectory_prompts (former prompts in the path) will be inserted into "state_transit" template and input into the optimizer to get "steps_per_gradient" new prompts.
> >
> > (d) **Q1: Clarifying humans are bad at managing multiple errors.** Thanks for asking this excellent question. The argument that "humans are bad at managing multiple errors" is mainly used against the limited working memory of humans, when iteratively polishing prompts based on observed errors. A human user might be able to iterate the trial-and-error step by considering one or two errors each time, but cannot handle many errors at the same time. More importantly, when sequentially performing the trial-and-errors, humans tend to polish the prompt locally at each step, without a global view of considering multiple errors strategically. This is similar to the MC-single baseline in RTable 6, with a directionless search process. A more accurate argument might be, "humans are bad at managing multiple errors at the same time." We will illustrate this clearly in the next version of the draft. Thanks!

---

> ### Author Response · Authors · 2023-11-22
> **Response to Reviewer j86n (Part 3)**
>
> (e) **Q2: Selecting hyperparameters.** Similar to many other empirical studies, we select hyperparameters based on intuitions and preliminary experiments. We now show additional hyperparameter selection experiments on iteration number and the exploration weight w_exp.
>
> RTable 7: Selecting hyperparameter: Iteration number
> | Iteration | 8 | 12 | 16 |
> | -------- | -------- | -------- | -------- |
> | CB     | 0.804    | 0.911    | 0.893   |
> | Biosses     | 0.625   | 0.750    | 0.725   |
> | Penguins     | 0.722    | 0.873    | 0.835   |
>
> **Iteration.** In the paper, we select the iteration based on our experience that a small number of iterations can't explore the prompt space well, but too many iterations are not necessary because they may have a potential problem of overfitting on the training set. We add new experiments in RTable 7 to test various choices of the iteration number, including iterations of 8, 12, and 16. The iteration of 8 is too small, which leads to a drop in performance, and the iteration of 16 is too large for the overfitting issue, leading to a marginal drop.
>
> RTable 8: Selecting hyperparamrter: Weight for balancing exploration vs. exploitation
> | w_exp | 1.0 | 2.5 | 4.0 |
> | -------- | -------- | -------- | -------- |
> | CB     | 0.768 (27 nodes)    | 0.911 (45 nodes)    | 0.857 (66 nodes)    |
> | Biosses     | 0.650 (30 nodes)    | 0.750 (36 nodes)    | 0.700 (55 nodes)     |
> | Penguins     | 0.759 (27 nodes)    | 0.873 (52 nodes)    | 0.835 (75 nodes)    |
>
>
>
> **w_exp.** In the paper, we choose w_exp=2.5 because the formula of the exploration rate is $w\_{exp}\times\sqrt{ln (N_{parent}/N_{child})}$, so we set some values of the visited times of parent node $N_{parent}$ and child node $N_{child}$. We find w_exp=2.5 can make the change of this value at a reasonable scale, when the $N_{parent}$ and $N_{child}$ increase. We also show new experiments in RTable 8, showing that when w_exp is small (1.0), the MCTS will only have one path, while w_exp is large (4.0), the MCTS will almost generate a new path in each iteration. Both of them are out of balance; w_exp is small, so the prompt space is not well explored; while w_exp is large, each path won't be visited multiple times to update the reward; we thus use w_exp=2.5 as a reasonable setting.
>
>
> (f) **Response to the ethics concerns.** We thank a lot for reviewer j86n for making such a great and thoughtful comment on being cautious about incorporating LLM feedback into the prompt. We couldn't agree more with adding an additional layer on top of prompting sensitive tasks, such as the loan approval task. The good news is that humans, especially domain experts, understand natural language prompts crafted by PromptAgent. Additional human supervision could be necessary sometimes, while we can further improve the method to be capable of detecting such biases and assigning lower rewards to these feedbacks. Specifically, PromptAgent presents a flexible framework to incorporate more powerful and advanced rewards to guide the prompt optimization process in a more principled way. Future rewards to measure the hallucinations, biases, ethical concerns, etc., of LLM feedback are very promising research directions to study.
>
> **References**:
> [1] Zhou, Y., Muresanu, A. I., Han, Z., Paster, K., Pitis, S., Chan, H., & Ba, J. (2022, September). Large Language Models are Human-Level Prompt Engineers. In The Eleventh International Conference on Learning Representations.

---

### Official Review · Reviewer_zmiy · 2023-10-29

**Soundness:** 2 fair
**Presentation:** 3 good
**Contribution:** 1 poor
**Rating:** 3
**Confidence:** 4

**Summary:**

This work introduces a prompting agent designed to perform few-shot prompting. The seeking of better-prompting policies is based on recursively generation and self-reflection (using a stronger general-purpose LLM). The idea is straightforward, the results show some improvement over single-round prompting methods.
The idea is interesting, but not technically novel, and the results are not insightful enough to add knowledge to the community (please see the weakness section below). I hereby would vote for a rejection.

**Strengths:**

The idea of this work is clear and easy to follow. The writing is in general clear. This idea can be useful from the engineering/ deployment side.

**Weaknesses:**

Technical contribution is limited.

Comparing the performance of an average user with LLM in prompting is somewhat unfair. Also, even human experts will be posited under an unfair setting where LLMs can do multiple-round prompting.

Some of the experiment settings are suspicious to be unfair (please see questions below)

**Questions:**

Can the authors please provide the depth setting used in the Greedy baseline? It is too-sample efficient but performs poorly in Figure 3.(a). I also wonder what would the optimization burden be for each task compared to a DFS search. The beam search baseline implemented in the main text seems to be the BFS search. I would expect DFS to outperform BFS as it can integrate more of the LLMs’ ability of reflection and reasoning.

---

> ### Author Response · Authors · 2023-11-22
> **Response to Reviewer zmiy (Part 1)**
>
> We thank reviewer zmiy for acknowledging that our idea is interesting, our work is clear and easy to follow, our writing is clear, and PromptAgent could be useful for greater practical usage. The reviewer's main concerns lie in the technical novelty, one baseline of human prompt, and some experiment settings regarding our ablation study of search variants. We now thoroughly address all the concerns and questions as follows:
>
> (a) **Clarifying the technical contribution.** We'd like to kindly redirect the reviewer to our global response (a), where we thoroughly address the concerns on the technical novelty and core contribution of PromptAgent. We hope our response could help the reviewer better understand the contributions of PromptAgent.
>
> (b) **Clarifying the baseline of human prompts.** We understand the reviewer's concerns that the average user or domain experts can also be further augmented with multi-round iterations by humans themselves. However, we argue that this concern shouldn't be considered as the weakness of our work for the following reasons:
> - First, as described in the section 4.1 Baseline paragraph, we try our best to directly adopt the human prompts from their original datasets and references. For datasets without readily available original human prompts, we carefully craft the prompts based on task description, options, and purpose of the tasks. The original dataset creator and we, as the normal users, create those human prompts, reflecting a prompting level of average users, thus serving as **a legitimate and rightful baseline to compare with**.
> - Second, adopting such an average level of human prompting has been **a traditional baseline for prompt optimization literature**. For example, Zhout et al. (2022) [1] proposed APE (Automatic Prompt Engineer), a Monte Carlo search method, to compare with and outperform human-level prompts, the same as our human prompt baseline. Then, almost all follow-up prompt optimization works for APE continue leveraging the iterative fashion of multi-round prompting and compete with the generic human prompts. Based on this, this concern is not limited to and cannot be blamed on our own work.
> - Third, instead of considering the disadvantage of humans when comparing with prompt optimization methods in general (not limited to our own work) as unfair, we'd better consider the multi-round prompting of prompt optimization methods as their **advantage of machine efficiency**. The disadvantage of humans under such circumstances is exactly the motivation for prompt optimization methods to explore advanced search methods to craft better prompts automatically, even with the cost of multi-round promoting. As stated in the second introduction paragraph of our paper, combining the merits of human trial-and-error and the machine's efficiency in doing iterative prompting creates more advanced methods, like PromptAgent.
> - Last but not least, human prompting is only one of our basic baselines, probably the weakest one in some tasks. We thoroughly consider **many other stronger baselines**, including human-written few-shot prompts, Chain-of-thought prompts, GPT Agent, and APE--a strong prompt optimization method, which thoroughly demonstrates the effectiveness of our proposed method. Therefore, the concern around one minor baseline is better not to be considered as the major weakness of our work.

---

> > ### Author Response · Authors · 2023-11-22
> > **Response to Reviewer zmiy (Part 2)**
> >
> > (c\) **Q1: Clarifying experiment settings for greedy search baseline**. We thank the questions from reviewer zmiy regarding the experiment settings of one of our ablation studies. We now clarify the questions about greedy search and add new experiments as the reviewer requested. The experiment rigor is one of our most central principles, and we hope our response could clear the concerns on "suspicious" and "unfair" settings.
> >
> > As reviewer zmiy mentioned, our greedy search and beam search are implemented similarly to BFS search. The Greedy baseline setting in our current version is a depth of 9 in total. The first level has 9 nodes, and the others have 3 nodes, so there are 33 prompts explored in total, which explains its sample efficiency in Figure 3 (a). When there are fewer explored nodes in greedy search, its performance is expected to be suboptimal due to the limited exploration space. We hope this could answer the reviewer's question regarding the greedy baseline.
> >
> > Moreover, the reviewer may wonder what will happen if we increase the exploration space of greedy search. Especially, the Beam baseline has a depth of 8, beam width=3, and each depth samples 9 prompts, so there are 72 explored prompts in total. We thus add a new experiment for a larger greedy search, Greedy-L, which has a depth of 8 and a total searched prompt 72, the same as Beam. We present the new results of five datasets in the following table, showing with a larger exploration, Greedy-L indeed can increase the performance, comparable with Beam baseline.
> >
> > RTable 4: Greedy-L baseline with a larger exploration space
> > | Tasks |  Greedy (33) | Beam (72) |Greedy-L (72) | MCTS (36~78) |
> > |----------|--------|-------|-------|-------|
> > | Penguins | 0.722  | 0.823 | 0.810| 0.873 |
> > | Biosses  |  0.575  | 0.675 |0.700 | 0.750 |
> > | Geometry | 0.475  | 0.610 |0.545 | 0.670 |
> > | Causal   | 0.580  | 0.640 |0.660 | 0.670 |
> > | Subj     |  0.645  | 0.765 |0.778 | 0.806 |
> > | Average  |  0.609  | 0.697 |0.698 | 0.754 |
> >
> > RTable 4 shows the new ablation study of Greedy-L with the same exploration space as the Beam and PrompgAgent. Note that all search baselines share the same state transition and action generation as PromptAgent, and the major variance is the search algorithm. We can see with the same exploration efficiency as Beam (the second column), Greedy-L (the third column) can reach comparable performance with Beam, indicating the importance of exploration space for the Greedy baseline. This also explains why the greedy performs poorly when it's super sample efficient (the first column). Nonetheless, PromptAgent with MCTS outperforms both Beam and Greedy-L significantly, under a similar efficiency budget.
> >
> > We show the exploration efficiency (the number of explored nodes) in the header brackets and re-draw figure 3 (a). [The new exploration efficiency figure](https://drive.google.com/file/d/195REJu1rcTScP9DBqMUVRTTv_czQ3Gt6/view?usp=sharing) clearly shows that PromptAgent with MCTS outperforms all baselines and search variants (we ignore the Beam points in the new figure as they're largely overlapped with the Greedy-L baseline) regarding the balancing of exploration efficiency and performance (high efficiency with high performance, clustering in the top-left corner), where greedy baseline cannot maintain such a balance properly (either high efficiency and low performance, or low efficiency and high performance).

---

> > > ### Author Response · Authors · 2023-11-22
> > > **Response to Reviewer zmiy (Part 3)**
> > >
> > > (d) **New experiments on the DFS-based baseline**. We thank the reviewer for suggesting an interesting baseline, which is the DFS-based search variant. We understand the reviewer's intuition that DFS probably could explore deeper paths first with more reflection and longer reasoning.
> > >
> > > To verify this, we implemented a standard DFS algorithm within the PromptAgent framework, inspired by recent ToT-related research [2]. We keep the same computation/exploration budget as other baselines, with a max depth of 8 and a total prompt number of 72, i.e., it will expand nodes and travers to the max depth, then backtrack to upper nodes to do DFS recursively. As expected, the DFS baseline tends to explore deeper levels of the tree; specifically, we find that in the Penguins task, the DFS baseline has 44 nodes with a depth of 8, while there are only 9 in Beam. We present the results for this DFS baseline in the following table.
> > >
> > > RTable 5: New ablation study: DFS-based search variant baseline
> > > | Tasks |  DFS | Beam  |Greedy-L | MCTS  |
> > > |----------|--------|-------|-------|-------|
> > > | Penguins | 0.759 | 0.823 | 0.810 | 0.873 |
> > > | Biosses  |0.675 | 0.675 | 0.700 | 0.750 |
> > > | CB       |0.821 | 0.839 | 0.822 | 0.911 |
> > >
> > > RTable 5 shows the results in three tasks comparing DFS, Beam, and Greedy-L. Note that all these three baselines have the same exploration efficiency. We can see that with deeper exploration, DFS is mostly comparable with the other two baselines, and lower than MCTS. We now provide further analyses for potential explanations:
> > > - Since DFS prefers the deeper paths, it usually leads to an imbalanced tree structure after exploring the prompt space, probably an insufficient pattern for the exploration.
> > > - We present some DFS prompts in [this anonymous doc](https://docs.google.com/document/d/1gDJs6PiawLVBRNGMEocjdftVfqbA9WGcJL9zrv8boGY/edit?usp=sharing), where we do observe that deeper exploration injects more nuanced, but sometimes unnecessary, information from LLM's deeper reflection. However, there is a balance of nuance and generalization for the exploration. When there are too many nuances in deeper nodes, it becomes overwhelming for the base model to understand these nuances, potentially leading to the overfitting problem.
> > >
> > >
> > > Nonetheless, the DFS is just one of the search variants in our ablation study, and we clearly observe that all search variants are worse than MCTS because of the strategic planning framework of MCTS, allowing it strategically search the vast prompt space to reach a proper balance of exploration (efficiency) and exploitation (performance).
> > >
> > > **References**:
> > >
> > > [1] Zhou, Y., Muresanu, A. I., Han, Z., Paster, K., Pitis, S., Chan, H., & Ba, J. (2022, September). Large Language Models are Human-Level Prompt Engineers. In The Eleventh International Conference on Learning Representations.
> > >
> > > [2] Yao, S., Yu, D., Zhao, J., Shafran, I., Griffiths, T. L., Cao, Y., & Narasimhan, K. (2023). Tree of thoughts: Deliberate problem solving with large language models. arXiv preprint arXiv:2305.10601.

---

### Official Review · Reviewer_WJJm · 2023-11-02

**Soundness:** 4 excellent
**Presentation:** 3 good
**Contribution:** 3 good
**Rating:** 8
**Confidence:** 4

**Summary:**

This paper proposes PromptAgent, which adopts a planning strategy based on MCTS for prompt engineering. Empirically, PromptAgent outperforms prior methods and human prompts. Overall the reviewer thinks the manuscript is well written and solid, and would like to recommend for acceptance.

**Strengths:**

1. This paper is well-written and easy to follow.
2. The method seems clean, straightforward, and promising.

**Weaknesses:**

There are several clarity issues in the experimental section regarding human prompts and reward functions. See questions below.

**Questions:**

1. At the end of Section 3.1, the manuscript says “PromptAgent straightforwardly defines a reward function $r_t=r(s_t,a_t)$ as the performance on a held-out set separated from the given training samples.” However, the reviewer cannot see how the reward functions are actually defined in the experimental sections (and the appendix). Is it possible that the author can provide a clear definition of how the reward function is defined (or provide an example of how the reward function is generated)?
2. In the paragraph “Baselines” of section 4.1, the descriptions of how Human Prompts are created are a bit vague. Although the authors have provided several examples of the human prompts in Appendix F, the reviewer would suggest the authors provide some extra details on how the human prompts are collected or generated.

---

> ### Author Response · Authors · 2023-11-22
> **Response to Reviewer WJJm**
>
> We are grateful to the Reviewer WJJm for recommending the acceptance of our paper and acknowledging that our work is solid and excellent sound, our writing is well-written and easy to follow, and our method is clean, straightforward, and promising. There are a few clarification questions from the reviewer, and we address them thoroughly as follows:
>
> (a) **Clarifying the reward function.** A clear definition of the reward function can be found on page 15 of our paper, "The reward function is determined by the base model's accuracy (or F1 score) on the sub-training set". To be more specific, for each task, we will split a part of the training set for reward calculation. For now, we use a simple strategy: calculate the accuracy or F1 score on the sub-training set. Moreover, similar to reward engineering in RL research, we expect future advanced rewards to be explored and crafted for PromptAgent, such as the self-evaluation and self-consistency rewards. Nonetheless, we found the task-specific reward works very well in the current PromptAgent, and we'll make its definition clearer in the main text of the next version.
>
> (b) **Clarifying human prompt baseline.** We thank the reviewer for digging into the details of the human prompt baseline. We surely will provide more details of this baseline in the new draft. Sepficially, in our paper, human prompts refer to human-written zero-shot prompts. For BigBench datasets[1], we follow previous works [2, 4] and directly utilize the task "description" provided in each dataset (One example can be found in [penguins_in_a_table](https://github.com/google/BIG-bench/blob/6436ed17f979b138463224421f8e8977b89076ed/bigbench/benchmark_tasks/penguins_in_a_table/task.json#L4)). These prompts are crafted and used by the dataset creators to instructively describe the dataset and prompt LLMs in their original paper [1]. These descriptions are also employed as the zero-shot baseline in the CoT evaluation paper by Suzgun et al. (2022) [2]. For the NCBI dataset, we reuse the prompt that was written by Gutierrez et al.[3], whose work explored the in-context learning ability of GPT-3 on the NCBI task. For other tasks, since there are no existing original prompts for us to reuse, we manually crafted the prompts based on the description, options, and purpose of the tasks. We craft the prompts to describe the task and options as normal users. We present all human prompts in Appendix F of the paper.
>
> **References**:
>
> [1] Srivastava, A., Rastogi, A., Rao, A., Shoeb, A. A. M., Abid, A., Fisch, A., ... & Wang, G. X. (2023). Beyond the Imitation Game: Quantifying and extrapolating the capabilities of language models. Transactions on Machine Learning Research.
>
> [2] Suzgun, M., Scales, N., Schärli, N., Gehrmann, S., Tay, Y., Chung, H. W., ... & Wei, J. (2022). Challenging big-bench tasks and whether chain-of-thought can solve them. arXiv preprint arXiv:2210.09261.
>
> [3] Gutiérrez, B. J., McNeal, N., Washington, C., Chen, Y., Li, L., Sun, H., & Su, Y. (2022, December). Thinking about GPT-3 In-Context Learning for Biomedical IE? Think Again. In Findings of the Association for Computational Linguistics: EMNLP 2022 (pp. 4497-4512).
>
> [4] Zhou, Y., Muresanu, A. I., Han, Z., Paster, K., Pitis, S., Chan, H., & Ba, J. (2022, September). Large Language Models are Human-Level Prompt Engineers. In The Eleventh International Conference on Learning Representations.

---

> > ### Comment · Reviewer_WJJm · 2023-11-22
> >
> > Thank you for your update! The reviewer has no further concerns now.

---

### Official Review · Reviewer_gPc5 · 2023-11-09

**Soundness:** 2 fair
**Presentation:** 3 good
**Contribution:** 3 good
**Rating:** 6
**Confidence:** 2

**Summary:**

The manuscript proposed PromptAgent, a new method using a planning algorithm, i.e., Monte Carlo Tree Search, to navigate and discover high-quality prompts through a process resembling human-like trial-and-error, incorporating feedback from model errors and refining previous prompts based on feedback. This method has been tested across 12 tasks with promising performance compared to existing baselines such as CoT and APE with GPT-3.5. The optimized prompt can be generalized to different LLMs, including GPT-4 and PaLM2.

**Strengths:**

- The proposed PromptAgent leveraged a Monte Carlo Tree Search framework to utilize errors and feedback identified by LLMs for the iterative refinement of prompts. This approach is theoretically sound and can enhance navigation through the expansive search space of potential prompts.
- PromptAgent showed promising experimental results across 12 tasks, and the optimized prompt can be generalized to different LLMs.
- PromptAgent showed better performance and exploration efficiency than other prompt optimization methods, including Automatic Prompt Engineer (APE).

**Weaknesses:**

- PromptAgent relies on a key hypothesis: the optimizer LLM (GPT-4 in this study) possesses adequate domain knowledge to identify the errors in the response from the base LLM and give meaningful feedback. However, this may not be a valid hypothesis, especially in some specialized areas such as medicine [1,2], where the data is relatively sparse due to strict data protection regularization like HIPAA.
- In order to refine the prompt, PromptAgent needs to concatenate the "error_string", "error_summarization and "trajectory_prompts" as one input. Challenges may arise in tasks demanding the interpretation of extensive contexts, such as the analysis of detailed medical documents, where the "state_transit" could become prohibitively large due to the number of training examples and the depth of the Monte Carlo Tree Search, potentially diminishing the LLMs' performance.

Reference

[1] Bhayana, R., Krishna, S., & Bleakney, R. R. (2023). Performance of ChatGPT on a radiology board-style examination: Insights into current strengths and limitations. Radiology, 230582.

[2] Azizi, Z., Alipour, P., Gomez, S., Broadwin, C., Islam, S., Sarraju, A., ... & Rodriguez, F. (2023). Evaluating Recommendations About Atrial Fibrillation for Patients and Clinicians Obtained From Chat-Based Artificial Intelligence Algorithms. Circulation: Arrhythmia and Electrophysiology, e012015.

**Questions:**

- If the optimizer LLM misidentifies an error or provides incorrect feedback, will this misinformation be propagated through the optimization process, leading to less effective prompts?
- This study used GPT-3.5 as the base model and a more capable model, such as GPT-4, as the optimizer LLM. Why not use GPT-4 for both base and optimizer LLM?
- The question above also extend to the implications of using fundamentally different LLMs, such as employing PaLM 2 as the base model against GPT-4 as the optimizer, and how this difference might affect the optimization outcome.
- In Fig 3(a), why GPT Agent, an "LLM-powered autonomous agent", was not compared in the exploration efficiency test?

---

> ### Author Response · Authors · 2023-11-22
> **Response to Reviewer gPc5 (Part 1)**
>
> We thank the reviewer gPc5 for acknowledging that our proposed method is theoretically sound to enhance navigation through the expansive search space of potential prompts, and our experimental results are promising across 12 tasks with better performance and exploration efficiency than other prompt optimization methods; moreover, recognizing the importance that the optimized prompt can be generalized to different LLMs. We especially appreciate the reviewer's deep insights into the domain knowledge of optimizer LLM and the potential issue of increasing the length of meta-prompts. We now thoroughly address all the concerns and questions as follows.
>
> **(a) Domain knowledge in the optimizer LLM.** We appreciate that the reviewer recognizes the importance of domain knowledge in our framework. We understand the reviewer's hypothesis regarding the necessity of domain knowledge in the optimizer LLM for providing insightful feedback. However, there are a few nuances we need to clarify, after which we will see the lack of domain knowledge for the optimizer LLM is not the problem this paper tries to solve, and actually, our PromptAgent framework can alleviate this burden of optimizer LLM to some degree.
>
> 1. First, we wish to clarify that PromptAgent with its current configuration (e.g., gpt-4 as the optimizer) is not posited as a universal solution for all domains, especially those where GPT-4's exposure is limited. Our primary focus has been on demonstrating the success and efficacy of PromptAgent across a broad range of tasks in BIG-Bench, specialized, and general NLP domains, where GPT-4's competency is already established [1].
>
> 2. Second, finding the suitable optimizer for a highly specialized domain (like medicine) is indeed an intriguing research area. This includes exploring methods to quantify the required domain knowledge for the optimizer and potentially training or fine-tuning domain-specific LLMs using specialized datasets, such as research papers or experiment data. Moreover, how to fine-tune an LLM under regularization like HIPAA is related to privacy-preserving research. However, such an exploration is beyond the scope of this paper and presents a promising avenue for future research. Importantly, the PromptAgent framework is designed to plug-and-play different optimizer models, allowing users to select/switch the most suitable one for their specific domain.
>
> 3. Third, the effectiveness of PromptAgent in deriving domain insights is not solely dependent on the optimizer LLM. It's a synergistic outcome of both the optimizer and our strategic planning framework. Specifically, we directly feed errors of the base model to the optimizer, saving the burden of the optimizer to detect these errors by itself. Moreover, the strategic planning framework will automatically select the high-reward path collecting meaningful error feedbacks, which alleviates the burden of the optimizer to produce "100%" accurate domain insights all the time (i.e., less accurate or wrong domain knowledge will be abandoned durig the search. A similar theme can be found in the following response (c\) to **Q1**). To illustrate this, we conducted new experiments replacing GPT-4 with GPT-3.5, a weaker LLM with much less domain knowledge. We'd expect a performance drop with the weaker optimizer. However, the following table shows that the drop is only marginal, indicating that even with a weaker optimizer, PromptAgent effectively improves performance across various tasks, including medical ones.
>
> RTable 1: Ablation study: Using both GPT-3.5 as base and optimizer LLM (second row)
> | Optimizer | NCBI (F1) | Biosses (Acc.) | Subjective (Acc.) |
> | -------- | -------- | -------- | -------- |
> | Initial Prompt | 0.521     | 0.550     | 0.517 |
> | GPT-3.5     | 0.615     | 0.675     | 0.747 |
> | GPT-4     | 0.645     | 0.750     | 0.806 |
>
> RTable 1 shows the results on two biomedical tasks and one general NLU task of ablating the optimizer LLM with GPT-3.5 and GPT-4. Note that the default base model is GPT-3.5. The first row is directly prompting GPT-3.5 with the initial prompt. The second and third rows use GPT-3.5 and GPT-4 for the optimizer LLM, respectively. As we can see, when replacing GPT-4 with GPT-3.5, containing much less domain knowledge, we still obtain substantial gain over the initial prompt baseline. While the performance is lower than GPT-4 as expected, the drop is tolerable and marginal, especially in the domain-specific task, NCBI, with only about $3$ point drop. Such results indicate the robustness of the overall PromptAgent framework to strategically search for high-quality domain insights, instead of solely relying on the knowledge of optimizer LLM.

---

> ### Author Response · Authors · 2023-11-22
> **Response to Reviewer gPc5 (Part 2)**
>
> In light of these clarifications, we hope to establish a mutual understanding with the reviewer that the limitation of the optimizer's domain knowledge, especially in highly specialized areas, is better not to be viewed as a weakness of this work. Instead, it represents a promising direction for future research when applying PromptAgent to solve more challenging domain problems. Whenever a prominent optimizer LLM is available, PromptAgent can make it plug-and-play, and leverage the strategic planning to help select and filter out/distill useful knowledge from it.
>
> **(b) Lengths of meta-prompts**. We thank the reviewer for carefully checking the details of our meta-prompts and spotting the increase in lengths that potentially impact the performance, especially for medical document tasks. To address this concern, we will show (i) our current lengths (even with the lengthy medical QA task) are within the manageable context length of modern LLMs, and more importantly, the growth of their lengths is also within a manageable speed, (ii) there are several tricks to reduce the meta-prompt length without hurting the performance too much. We make a few clarifications and further analyses as follows:
>
> 1. First, the context length problem of LLMs has received an extensive amount of attention recently, from 512, 1024, and 2048 to 8k, 16k, 32k, and even 100k (just put a [reference link](https://cobusgreyling.medium.com/rag-llm-context-size-6728a2f44beb) here). While some recent research studied the [lost-in-the-middle issue](https://arxiv.org/abs/2307.03172), the capacity of long context understanding has been greatly improved. Then, similar to the above response about the choice of the optimizer LLM, whenever another LLM with a longer context is more suitable for our downstream tasks, such as medical document tasks, PromptAgent can combine with it in a plug-and-play fashion. In other words, with the increase in modern LLM's context length, we anticipate the increase in prompt lengths won't be a fatal issue for PromptAgent.
>
> 2. Second, we carefully analyze the length statistics in PromptAgent. Specifically, we focus on the med_qa task, which contains relatively long documents compared with other tasks (question length ranges from 400 to 1200), similar to the medical document tasks suggested by the reviewer. [This anonymous link](https://drive.google.com/file/d/19ICrn7cOPRuaIrTXDB_nb_KNZFxM81a0/view?usp=sharing) shows a figure for the growth of state_transit's average lengths, w.r.t the depth. With the maximum length of 4k for this length task, we can assure that it can be well handled by GPT-4 with 8k context length.
>
> 3. Moreover, from the [figure](https://drive.google.com/file/d/19ICrn7cOPRuaIrTXDB_nb_KNZFxM81a0/view?usp=sharing), we observe the relatively linear growth pattern, suggesting the growth of "state_transit" won't lead to prohibitive long prompt, even with the increase of depth. Note that the number of training examples fed to each state is bounded by the batch size, which is fixed during the search. To confirm this, we further investigate the contributing factors for the length increase of "state_transit". As clearly illustrated in Table 7 from the paper, there are two parts of "state_transit": error information consisting of "error_string" and "error_summarization", and history information with "trajectory_prompts". "error_string" includes the current task prompt, question, and model's output. As the depth goes deeper, e.g., increase 1, the task prompt could grow; also, one more task prompt is added to the "trajectory_prompts". Therefore, the main contributing factor to the length increase of "state_transit" is the growth of the task prompt, which grows slowly and even steadily during the search process.
>
> 4. Finally, we did new experiments studying two ways to reduce the length of "state_transit." (i) Use a smaller batch size: batch_size=1 and (ii) Remove error_string and use fewer prompts in "trajectory_prompts". [This new length analysis figure with batch_size=1](https://drive.google.com/file/d/16w2LiMK7LjVAGXvnxDUwR8XDezbgsv4M/view?usp=sharing) shows we can reduce the med_qa's maximum length from 4000 to 2400, and [another figure shows removing error_string with less prompts in trajectory_prompts](https://drive.google.com/file/d/1_YHS23zpz5w78MF_MMMU32YQIWuQmfWN/view?usp=sharing) reduces the max prompt length from 4000 to 1600. Moreover, we show their performance in the following table, RTable 2.

---

> ### Author Response · Authors · 2023-11-22
> **Response to Reviewer gPc5 (Part 3)**
>
> RTable 2: Ablation study on reducing "state_transit" length
> |  | NCBI (F1)  | Med_qa (Acc.) |
> | --------  | -------- | -------- |
> | Initial Prompt | 0.521     |  0.508 |
> | Short state_transit     | 0.591    | 0.554 |
> | batchsize=1 | 0.627     | 0.600 |
> | batchsize=5     | 0.645     |  0.570 |
>
> RTable 2 shows the performance of two remedies for reducing the meta-prompt length on two biomedical tasks, which usually come with longer input questions. As we can see from the figure, both remedies can effectively reduce the meta-prompt lengths, with a marginal performance drop. Moreover, reducing the batch_size seems to be more effective in reducing the length while maintaining the performance than deleting "error_string" to shorten the meta-prompt. Interestingly, for the med_qa task with batch_size=1, we observe a slightly better performance than our default batch_size=5. This implies that for lengthy tasks, a shorter meta-prompt containing one single long example could potentially lead to a better performance. Nevertheless, we show that there are effective strategies to reduce the meta-prompt lengths while maintaining the performance.
>
>
> **(c\) Q1: Influence of low-quality error feedback.** It is possible for the optimizer to produce less accurate error feedback in one optimization step, which results in less effective child nodes. However, this is exactly why we need strategic planning to help explore the complex expert-prompt space. The ability of strategic planning allows PromptAgent to simulate future rewards and backtrack from low-quality error feedbacks. More specifically, the quality of the error feedback will be measured by the reward, and PromptAgent will avoid the low-quality error feedback with a low reward to explore in future iterations. Therefore, the low-quality error feedback will have less impact on the overall performance.
>
> **(d) Q2: Clarifying why not use GPT-4 for both base and optimizer LLM.** Using both GPT-4 for both base and optimizer LLM is expected to reach higher performance. Our generalization experiments in Table 3 and Figure 1 (b) indirectly demonstrate this. Specifically, the prompts used for generalization experiments use GPT-3.5 as the base model and GPT-4 as the optimizer. When transferred to GPT-4 as the base model, these prompts lead to higher performance. Therefore, we anticipate a similar performance increase if we directly use GPT-4 both for base and optimizer LLMs. However, the main reason that prevents us from using GPT-4 for both base and optimizer LLM is the expensive cost of GPT-4, which is 40 times more expensive than GPT-3.5, when we were conducting experiments. Running our PromptAgent in a small dataset, like penguins in the table, could cost `$`20 for the base model portion, and will cost `$`800 if using GPT-4 as the base model, which is unaffordable for us. Nonetheless, our generalization experiments show promise by transferring GPT-3.5-based prompts to GPT-4, saving the high cost of directly using GPT-4 as the base model.
>
> **(e) Q3: New experiments in using PaLM 2 as the base model.** As stated in the above response, the choice of the base model is very flexible in the PromptAgent framework, which could be GPT-3.5, GPT-4, PaLM, or other LLMs. However, using a weaker LLM as the base model, such as PaLM 2 (chat-bison-001), could lead to a performance drop, partially due to the weaker understanding of expert prompts. This can be indirectly observed from the generalization experiment (Table 3 and Figure 1 (b)), where we transfer GPT-3.5-based prompts to PaLM 2 with a performance drop. We add new experiments in the following table to directly observe the effect of using PaLM 2 as the base model. As we mentioned in the paper, PaLM 2 is weaker than GPT-3.5, but we can see PromptAgent can still improve its performance greatly.
>
> RTable 3: Ablation study: using PaLM 2 as the base model
> | Base Model | Penguins Acc. | Biosses Acc. | CB Acc. |
> | -------- | -------- | -------- | -------- |
> | Initial Prompt (PaLM2)|  0.215    | 0.500     |0.571  |
> | Optimized Prompt (PaLM2) | 0.443     | 0.600     |0.839  |
> | Initial Prompt (GPT-3.5)     |0.595      | 0.550     |0.714  |
> | optimized Prompt (GPT-3.5)     | 0.873     | 0.750     | 0.911 |
>
> RTable 3 shows the results in three tasks using different base models, PaLM 2 and GPT-3.5, both against GPT-4 as the optimizer. The first row shows the worse initial performance using PaLM 2, indicating its weaker understanding of human prompts. After using PromptAgent to optimize PaLM 2 in the second row, we observe significant improvement over the first row, indicating the effect of PromptAgent to optimize weaker LLMs.

---

> > ### Author Response · Authors · 2023-11-22
> > **Response to Reviewer gPc5 (Part 4)**
> >
> > (f) **Q4: Clarifying the exploration efficiency for the GPT Agent baseline.** We would like to further clarify the baseline of the GPT Agent. The GPT Agent we used in the paper was based on the web platform (https://chat.openai.com/) using the GPT-4 plugin. It didn't use search algorithms, and we iteratively fed task examples to the agent by humans, letting the agent plugin optimize the prompts by itself. Therefore, it is not applicable to compare search efficiency with our method.

---

> > > ### Comment · Reviewer_gPc5 · 2023-11-23
> > > **Thanks for the responses**
> > >
> > > I am grateful for the comprehensive discussions and new experimental results presented in response to my earlier concerns. Particularly, Response (b) offers valuable insights into the scalability of the proposed method.
> > >
> > > For Response (a), I appreciate the authors clarifying that "PromptAgent with its current configuration (e.g., gpt-4 as the optimizer) is not posited as a universal solution for all domains, especially those where GPT-4's exposure is limited." and further discussed the effectiveness of PromptAgent particularly when the optimizer LLM potentially lacks the domain knowledge. A recent study [1] discussed the related topic and may be incorporated to further enhance this discussion.
> > >
> > > However, I cannot agree with the authors' claim regarding "plug-and-play." This is true for PromptAgent, but also for many other prompting/optimizing strategies. It is not practical to just "plug-and-play" considering the cost and time, especially when an advanced LLM (e.g. GPT-4) is employed (as mentioned in Response (d)). A more detailed discussion about these potential limitations would be beneficial for readers.
> > >
> > > Therefore, I strongly recommend that the authors incorporate the responses into a thorough discussion of the limitations and future work of this research in the revised manuscript (please correct me if I am missing it). This aspect, currently absent from the submission, is crucial for providing a well-rounded understanding of the work.
> > >
> > > In summary, while acknowledging the improvements and detailed responses, **my initial rating remains unchanged**. And I strongly recommend the inclusion of a detailed section on limitations and future work in the revised manuscript.
> > >
> > > Reference
> > >
> > > [1] Yin, Z., Sun, Q., Guo, Q., Wu, J., Qiu, X., & Huang, X. (2023). Do Large Language Models Know What They Don't Know?. arXiv preprint arXiv:2305.18153.

---

> > > > ### Author Response · Authors · 2023-11-23
> > > > **Follow-up Response to Reviewer gPc5 (part 1)**
> > > >
> > > > We are so grateful to reviewer gPc5 for the quick response to our rebuttal, and we're encouraged to see reviewer gPc5 is satisfied with our "comprehensive discussions" and "new experiment results," especially "valuable insights into the scalability of our proposed method." Rest assured, these insights will be thoughtfully integrated into the next version of our manuscript.
> > > >
> > > > In addressing the concerns about [domain knowledge within the optimizer LLM](https://openreview.net/forum?id=22pyNMuIoa&noteId=3oyWA85fGv), reviewer gPc5 and we are having an inspiring and engaged discussion here, pushing us really hard to think of the limitations of LLMs (with a great reference provided [1]) and PromptAgent in solving real challenging domain problems.
> > > >
> > > > First of all, we're glad that both reviewer gPc5 and we agree that "the current configuration of PromptAgent is not posited as a universal solution for all domains," and perhaps, such a universal solution is very likely doesn't exist at all. Moreover, the scope of knowledge within contemporary models like GPT-4 is an area of active research [1, 3, 4], and its limitations are more a reflection of the current state of LLMs than a specific shortcoming of our PromptAgent framework.
> > > >
> > > > Then, the only remaining concern from the reviewer gPc5 seems to be the practical solution/idea for the question, "How to adopt PromptAgent to a highly specialized domain when the domain knowledge of the optimizer is limited." For example, reviewer gPc5 challenges the "plug-and-play" solution, for it ignores the domain and problem nuances, such as cost and time for running LLMs. We now provide a comprehensive discussion on potential solutions (of course, each comes with its own limitations):
> > > > - So far, our current work has proven to be successful by adopting GPT-4 as the optimizer to solve a wide range of tasks, including table understanding, geometry, temporal understanding, counting, natural language understanding, biomedical tasks, etc. The cost and time for using GPT-4 as the optimizer is tolerable in our settings.
> > > > - **Direct use of GPT-4 for specialized domains**: one straightforward solution would be to directly use GPT-4 for many specialized domains, potentially with specialized knowledge prompts to trigger certain domain knowledge. There are certain domains that leak very little information to the pre-training corpus of GPT-4, possibly due to the data protection regularization (but we really don't know what data GPT-4 has been pre-trained with). Given the impressive generalizability of LLMs, GPT-4 has shown promises to lots of scientific domains that were thought to be hard for LLMs, such as clinical domain [2, 3], medicine [5], scientific domains like drug discovery, biology, computational chemistry, materials design, etc. [4]. Considering the third argument in our response (a) to reviewer gPc5, that PromptAgent can alleviate the burden of the optimizer's domain knowledge, we would anticipate the reasonable performance of directly applying PromptAgent with GPT-4 as the optimizer to many specialized domains. Note that using GPT-4 as the optimizer doesn't pose a very high cost for API usage (a few error feedback generation and state transition promptings), and the inference time is bounded by the rate limit of OpenAI accounts.

---

> ### Author Response · Authors · 2023-11-23
> **Follow-up Response to Reviewer gPc5 (part 2)**
>
> - **Alternative strategies for domain knowledge enhancement**: When the domain knowledge of GPT-4 is proven to be inadequate or the cost/time/privacy of GPT-4 is less acceptable, we then need to resort to open-sourced models (fine-tuning GPT models via OpenAI APIs is possible but seems to be less preferred generally due to their high cost) for either pre-training from scratch or fine-tining an existing prominent one like Llama 2 on domain-specific data including research paper, experiment records, etc. If that's the case, the "plug-and-play" solution might lead to potential problems, such as generating incoherent sentences or very low-quality error feedbacks, etc. We then need to design novel solutions on top of PromptAgent to overcome the knowledge limitation of the optimizer. We list several potential directions as follows:
>     - Using **retrieval techniques** to search for reliable and faithful domain knowledge directly to improve the error feedback
>     - Employing **an additional layer of quality assessment** for error feedbacks before incorporating them into the optimization step to ensure their coherence and relevance
>     - Proposing **a hybrid of optimizers**, combining the general capabilities of LLMs with domain-specific LLMs to balance the broad knowledge of general LLMs with targeted expertise
>     - **Collaborating with humans**: adaptively ask domain experts to provide initial guidance to shape the direction of the optimization; while this could be expansive, it can towards more relevant and accurate domain insights
>     - **Data augmentation for domain expertise**: for domains where privacy and data sensitivity are paramount (like healthcare), fine-tuning LLMs using synthetic or anonymized data can be a path forward
>
>
> In conclusion, we recognize that the application of PromptAgent to highly specialized domains presents unique challenges. However, we believe that the strategies outlined above, combined with the inherent adaptability of the PromptAgent framework, provide a strong foundation for future research and practical applications in this area. We are committed to further exploring these solutions and sharing our findings with the community.
>
> As suggested by reviewer gPc5, **we have incorporated our great discussion of the limitation in the updated manuscript (Appendix A)**. We hope this additional discussion addresses the concerns raised and illustrates our dedication to the continuous improvement and application of PromptAgent in a variety of challenging domains.
>
> Thank you again for your insightful feedback and for pushing us to think critically about the potential and limitations of our work.
>
>
>
> **References:**
>
> [1] Yin, Z., Sun, Q., Guo, Q., Wu, J., Qiu, X., & Huang, X. (2023). Do Large Language Models Know What They Don't Know?. arXiv preprint arXiv:2305.18153.
>
> [2] Yao, Z., Jaafar, A., Wang, B., Zhu, Y., Yang, Z., & Yu, H. (2023). Do Physicians Know How to Prompt? The Need for Automatic Prompt Optimization Help in Clinical Note Generation. arXiv preprint arXiv:2311.09684.
>
> [3] Hernandez, E., Mahajan, D., Wulff, J., Smith, M. J., Ziegler, Z., Nadler, D., ... & Alsentzer, E. (2023, June). Do We Still Need Clinical Language Models?. In Conference on Health, Inference, and Learning (pp. 578-597). PMLR.
>
> [4] AI4Science, M. R., & Quantum, M. A. (2023). The Impact of Large Language Models on Scientific Discovery: a Preliminary Study using GPT-4. arXiv preprint arXiv:2311.07361.
>
> [5] Thirunavukarasu, A. J., Ting, D. S. J., Elangovan, K., Gutierrez, L., Tan, T. F., & Ting, D. S. W. (2023). Large language models in medicine. Nature medicine, 29(8), 1930-1940.

---

### Author Response · Authors · 2023-11-22
**Global Responses and Summary of Authors' Response (Part 1)**

We would like to thank all the reviewers for their constructive feedback! We are encouraged to see that reviewers find: **(a)** our work is new and promising, and the idea is interesting (gPc5, zmiy, WJJm), **(b)** our proposed method is clean, clear, and easy to follow (WJJm, zmiy), **(c\)** the approach is solid, theoretically sound and can enhance navigation through the expansive search space of potential prompts (gPc5, WJJm, j86n), which could be **(d)**  useful from the engineering/ deployment side (zmiy). Also, **(e)** our presentation is good, clear, well-written, and easy to follow (gPc5, WJJm, zmiy, j86n), **(f)** our results across 12 tasks are promising and comprehensive, and the optimized prompt can be generalized to different LLMs (gPc5, j86n) with better performance and exploration efficiency than other prompt optimization methods (gPc5), where ablation studies shed further lights on the effectiveness of the proposed strategic exploration of the search space using MCTS (j86n).

We have addressed all the questions raised by reviewers with additional experiments and thorough clarifications via separate responses to each reviewer. Among them, we'd like to clarify/re-emphasize the core contributions of our PromptAgent in the global response (although most reviewers don't question our technical novelty too much).

(a) **Core technical novelty and contributions.** There are two crucial and novel contributions we'd like to highlight in the global response.

1. **A new prompting problem--expert-level prompt optimization**. Unlike discovering magic/local prompt variants as many existing prompt optimization methods are doing (e.g., discovering effective variants to "let's think step by step"), expert-level prompt optimization aims to craft prompts equivalent in quality to those handcrafted by experts, to save human effort in tedious prompt engineering. There are **three unique features** that characterize expert-level prompts: **(i)** capturing task nuances and high-quality structured instructions reflecting the depth of domain knowledge, **(ii)** interpretable and rationalizable by domain experts; and **(iii)** transferrable across various models and human experts, since good domain knowledge should be generalizable. As we anticipate the emergence of many larger LLMs that can understand intricate instructions, expert prompts will become more important to unleash their full potential (without fine-tuning their parameters). We are **the first** to propose and formally study this relatively new problem. This problem has started to receive attention recently, e.g., a recent work [2] released a few days ago studied how to craft long prompts to capture more nuances and domain insights as well.

2. The most critical technical novelty of PromptAgent is the **reformulation of prompt optimization problem**. Most existing works have traditionally treated prompt optimization as a diverse sampling problem, aiming to sample as diverse as possible prompt variants (e.g., via LLM sampling or evolutionary algorithms) without considering the balance of exploration (sample something new) and exploitation (sample too much). PromptAgent views this as a strategic planning problem similar to how humans navigate the vast prompt space (but augment the capacity of humans with greater machine efficiency). This new formulation enables a principled framework to maintain a proper balance of exploration and exploitation, and requires novel designs of state, action, and reward to make it work. We believe we're **the first** to propose this paradigm shift and present a principled framework to study prompt optimization by unifying prompt sampling, error reflection, and rewarding.

In addition to these two key highlights, we also have the following **novel empirical contributions** that could be of great interest to the LLM community:
- We successfully implement MCTS within this strategic planning framework, demonstrating its effectiveness across 12 tasks spanning three practical and challenging domains. Both extensive quantitative and qualitative results reveal the ability of PromptAgent to distill insightful domain knowledge from model errors in training data and the optimizer LLM's knowledge.
- We are **the first** among prompt optimization literature to show the transferability of optimized prompts across various base models. Different from optimized magic prompts towards each specific LLM, this generalizability is unique to expert prompts and is highly desirable, as mentioned above.
- We are **the first** to thoroughly study the problem of exploration efficiency in the prompt optimization literature. We want to promote this balance of exploration efficiency and performance as a new evaluation metric for prompt optimization literature (rather than keep chasing the performance). Better methods are ideal for having high exploration efficiency with higher performance.

---

> ### Author Response · Authors · 2023-11-22
> **Global Responses and Summary of Authors' Response (Part 2)**
>
> Given all the clarifications, we hope to address a few potential comments from different opinions on our contributions as follows:
> - "use of MCTS in prompt engineering is not novel": Interpreting our PromptAgent as "applying MCTS in prompt engineering" probably isn't the best way to understand its critical contribution/position in the literature.
>     - First, as illustrated above, the reconceptualization of prompt optimization comes first, then we have MCTS to operationalize this new formulation. Focusing solely on the technique used risks overshadowing the broader, more impactful aspect of our work, which is the novel framing of the problem itself first. This new perspective opens the door to a variety of strategic planning approaches for prompt optimization, with MCTS being just one of the potential solutions we have explored in our current research.
>     - Second, while MCTS has indeed been employed in prompting research, its application in prompt optimization is unprecedented, as detailed in our related work section. Moreover, PromptAgent's use of MCTS goes beyond a straightforward implementation. It includes significant innovations, including novel formulation of state/action/reward/meta-prompts to tackle specific challenges for prompt optimization, adapting the algorithm to the nuances of expert-level prompts, and balancing exploration and exploitation in a novel and complex domain, etc.
> - "recursive generation" and "self-reflection" are existing techniques, "not technically novel": It is true that some components in PromptAgent are not brand new concepts per se; the innovation of our work lies in the holistic integration of these elements into a larger, cohesive framework. This isn't simply a case of assembling existing parts; it's about orchestrating them in a way that specifically serves the greater goal of expert-level prompt optimization. PromptAgent redefines these components, enabling them to interact dynamically within a strategic planning paradigm—a concept previously unexplored in this field.
>
> To sum up, after all the above clarifications, we believe the concerns about the technical novelty of our work have been effectively resolved. We hope our responses highlight the significant contributions and potential of expert-level prompting. PromptAgent not only offers a novel approach to prompt optimization but also paves the way for a deeper understanding and utilization of large language models in complex tasks.
>
> Last but not least, we will add all of these additional experiments and suggestions to the updated PDF. Below, we summarize the main changes during the rebuttal and request the reviewers to take a look at the new additions. Please also be careful to check some **new figures** we provided through **anonymous links embedded in the text** (cannot directly embed them into openreview responses).
> - As suggested by gPc5: new experiment results on ablating the optimizer from GPT-4 to GPT-3.5, showing PromptAgent can help alleviate the high demand for the optimizer's domain knowledge
> - As suggested by gPc5: new experiment results on studying effective strategies in reducing the lengths of meta-prompts
> - As suggested by gPc5: new experiment results on ablating the base model from GPT-3.5 to PaLM 2
> - As suggested by gPc5: clarifying how PromptAgent can avoid low-quality error feedback
> - As suggested by gPc5: adding discussions about limitations and promising future works to deploy PromptAgent in highly specialized domains
> - As suggested by WJJm: clarifying details of the reward function and human prompt baseline
> - As suggested by zmiy: clarifying the legitimacy of human prompt baseline
> - As suggested by zmiy: new experiment results on greedy search baseline with a bigger exploration space
> - As suggested by zmiy: new experiment results on a DFS-based search variant baseline
> - As suggested by j86n: clarifying the necessity and contribution of MCTS (through our core ablation study on search variants); we implement a new baseline as error feedback-based APE
> - As suggested by j86n: new experience results on hyperparameter selection

---

### Meta-Review · Area_Chair_DAe1 · 2023-12-05

**Metareview:**

This paper presents a new way to optimize prompts using MCTS. While reviewers have concerns about the overall technical novelty, I appreciate that this is this the first to formalize, and analyze exploration efficiency in the search space and also find transferability of optimized prompts across various base models -- which is valuable. There was consensus that the paper is well written and experiments are solid.

**Justification For Why Not Higher Score:**

concerns about the overall technical novelty

**Justification For Why Not Lower Score:**

first to formalize, and analyze exploration efficiency in the search space and also find transferability of optimized prompts across various base models

---

### Decision · Program_Chairs · 2024-01-16

Accept (poster)